# Nitric oxide is a host cue for *Salmonella* Typhimurium systemic infection in mice

Lingyan Jiang [1,2], Wanwu Li[1,2], Xi Hou[1,2], Shuai Ma[1,2], Xinyue Wang[1,2], Xiaolin Yan[1,2], Bin Yang [1,2], Di Huang[1,2], Bin Liu[1,2] & Lu Feng [1,2 ✉]

Nitric oxide (NO) is produced as an innate immune response against microbial infections. *Salmonella* Typhimurium (*S.* Typhimurium), the major causative pathogen of human gastro-enteritis, induces more severe systemic disease in mice. However, host factors contributing to the difference in species-related virulence are unknown. Here, we report that host NO production promotes *S.* Typhimurium replication in mouse macrophages at the early infection stage by activating *Salmonella* pathogenicity island-2 (SPI-2). The NO signaling-induced SPI-2 activation is mediated by Fnr and PhoP/Q two-component system. NO significantly induced *fnr* transcription, while Fnr directly activated *phoP/Q* transcription. Mouse infection assays revealed a NO-dependent increase in bacterial burden in systemic organs during the initial days of infection, indicating an early contribution of host NO to virulence. This study reveals a host signaling-mediated virulence activation pathway in *S.* Typhimurium that contributes significantly to its systemic infection in mice, providing further insights into *Salmonella* pathogenesis and host–pathogen interaction.

[1] The Key Laboratory of Molecular Microbiology and Technology, Ministry of Education, Nankai University, Tianjin, China. [2] TEDA Institute of Biological Sciences and Biotechnology, Tianjin Key Laboratory of Microbial Functional Genomics, Nankai University, Tianjin, China. ✉email: fenglu63@nankai.edu.cn

nnate immune responses, including the production of reactive oxygen species (ROS) and reactive nitrogen species (RNS), represent the first line of host defense against microbial infections by limiting replication and spread of invading pathogens[1]. On the other hand, microbial pathogens have developed strategies to antagonize innate immune responses and some even exploit those responses as signals to promote their own virulence[2,3]. The interaction between host innate immune functions and pathogen virulence mechanisms largely determines the outcome of most infections[4].

*Salmonella* is an important intracellular pathogen that causes a range of diseases in humans and animals. Infections by *Salmonella* represent a considerable burden in both developing and developed countries with more than 110 million cases in humans reported every year[5,6]. Depending on serovar/host combinations, *Salmonella* infections are generally either localized to the gastrointestinal tract, resulting in mild gastroenteritis, or disseminated further to extraintestinal sites, causing severe systemic diseases such as typhoid fever[7–9]. *Salmonella enterica* serovar Typhimurium (*S.* Typhimurium) is the main causative agent for human gastroenteritis but can induce a typhoid-like systemic infection in mice[10,11]. How *S.* Typhimurium interacts differently with hosts, leading to the severe virulence in mice is not clear.

The ability to survive and replicate in host macrophages is essential to the systemic infection of *S.* Typhimurium[12,13]. Accordingly, *S.* Typhimurium replicates to high numbers in mouse macrophages, in a manner that requires a specific type III secretion system (T3SS) encoded by *Salmonella* pathogenicity island-2 (SPI-2)[14]. Expression of SPI-2 genes under the control of SPI-2-encoded SsrA (the integral membrane cognate sensor)/SsrB (the response regulator) two-component regulatory system (TCS), the master regulator of all SPI-2 operons, is induced upon phagocytosis by various host cues, including cation deprivation, phosphate starvation, and phagosome acidification generated by innate immune responses[15,16]. The SPI-2-encoded T3SS injects effector proteins, both encoded within and outside of SPI-2, into host cells to induce the formation of a specialized membrane-bound compartment called the *Salmonella*-containing vacuole (SCV) within which *S.* Typhimurium survives and replicates[17–19]. SPI-2 mutants of *S.* Typhimurium showed intracellular proliferation defect in primary mouse peritoneal macrophages (PMs) and mouse macrophage cell lines such as RAW264.7 and J774A.1[20–22], and were found to be highly attenuated in the systemic infection mouse model infected orally, intraperitoneally (i.p.), or intravenously[20,22–24]. Although the expression of SPI-2 is also induced in human macrophages, which are less permissive for *S.* Typhimurium replication, the induction level is much lower than that observed in mouse macrophages[25,26], indicating the presence of host-specific cues for SPI-2 activation in mouse macrophages.

Nitric oxide (NO), the RNS prototype, is produced as an innate immune response against microbial infection by inducible nitric oxide synthase (iNOS), which converts arginine into citrulline and NO[27,28]. While ROS is responsible for the initial killing of pathogens, RNS reportedly inhibits *S.* Typhimurium replication in infected organs and within infected macrophages at later stages[29,30]. However, NO by itself poses only weak antibacterial activity[31]. Noticeably, NO production was detected in mouse macrophages (5–20 μM) infected with *S.* Typhimurium[32,33]. However, human macrophages, which are less permissive for *S.* Typhimurium replication, generally show low levels of iNOS expression and produce negligible amounts of NO[34–37]. This suggests a positive correlation between host NO production and SPI-2 expression and *S.* Typhimurium replication. Interestingly, *S.* Typhimurium avoids direct contact with both ROS and RNS by residing within the SCV[33,38]. However, as NO is freely diffusible

into the SCV, it is curious whether host NO production contributes to severe *S.* Typhimurium virulence in mice.

TCSs are signaling pathways via that bacteria sense and respond to environmental or cellular parameters and thus adapt to the changing conditions[39]. The PhoP/Q TCS, a master regulator of *S.* Typhimurium virulence functions, generally activates SPI-2 via both *ssrB* at the transcriptional level and *ssrA* at the posttranscriptional level by sensing multiple host cues[40]. Intramacrophage signal sensing by PhoP/Q is critical for SPI-2-mediated virulence of *S.* Typhimurium in mice[40,41], as *S.* Typhimurium mutants lacking the PhoP/Q system are highly attenuated for virulence in mice and unable to survive within macrophages[12,42]. Whether PhoP/Q senses or responds to host NO production has not been investigated.

Fnr is a well-known global anaerobic regulator and contributes to the virulence of many bacterial pathogens that encounter changes in $O_2$ availability[43–45]. Fnr is also required for *S.* Typhimurium systemic infection, as the *fnr* mutant was completely attenuated in both orally and i.p.-infected mice and the lack of *fnr* resulted in a dramatic reduction in the ability of *S.* Typhimurium to replicate in PMs[46]. A microarray-based transcriptome analysis reveals that Fnr positively regulates many invasion-associated virulence genes, but has no effect on SPI-2, in *S.* Typhimurium[46]. As this study was conducted using *S.* Typhimurium grown in Luria–Bertani (LB) to log phase, which is a non-SPI-2-inducing condition[16], whether Fnr contributes to SPI-2 induction in host macrophages is uncertain. Fnr is also known as a NO-responsive regulator in bacteria[44]. Binding to NO inactivates Fnr by reacting with its $[4Fe-4S]^{2+}$ cluster, thereby impacting the expression of Fnr-regulated genes[45]. Whether NO affects the transcription level of *fnr* is not known.

In this study, the contribution of host NO production to *S.* Typhimurium systemic infection in mice was investigated using mouse infection assays, gentamicin protection assays, RNA sequencing (RNA-seq), SPI-2 gene (*ssrA*) promoter substitution analyses, and many other molecular techniques. We found that host NO production promotes *S.* Typhimurium replication in mouse macrophages by activating SPI-2, and increases the bacterial burden in the liver and spleen of infected mice at early infection stages. Further investigations revealed that the NO-signaling-induced SPI-2 activation is mediated by Fnr and PhoP/Q. Thus, this study reveals that NO produced by the innate immune system is a host cue for *S.* Typhimurium virulence activation in mice, providing further insight into *Salmonella* pathogenesis.

## Results

**Host NO levels correlate positively with *S.* Typhimurium replication and SPI-2 expression levels in mouse macrophages.** Previous studies reported that NO is produced in mouse but almost not in human macrophages in response to *S.* Typhimurium infection[32,33,47], and this was confirmed here. Up to 20 μM NO (within the range reported previously) was detected in mouse RAW264.7 while less than 1 μM was detected in human U937 macrophage cells during the 24 h infection period with *S.* Typhimurium wild-type strain ATCC 14028 s (Fig. 1a). Negligible NO production was also detected in human THP-1 cells (Supplementary Fig. 1a). The difference in NO production between RAW264.7 and U937 cells with LPS treatment was further confirmed (Supplementary Fig. 1b). Providing the different virulence to mice (high) and humans (low), *S.* Typhimurium replicated better in mouse macrophages than in human macrophages[48]. In agreement, intracellular bacterial burden of *S.* Typhimurium was much higher in RAW264.7 cells than that in U937 cells as indicated by gentamicin protection assays (Fig. 1b).

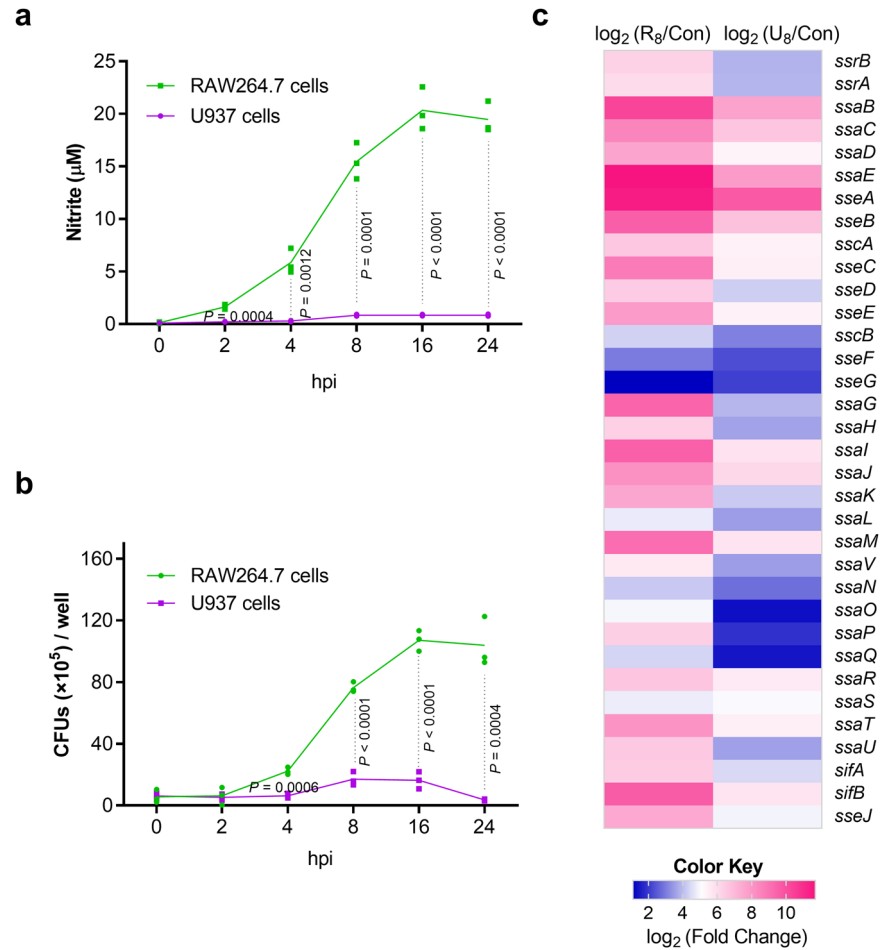

**Fig. 1 Host NO levels correlate positively with *S.* Typhimurium replication and SPI-2 expression levels in mouse macrophages. a** Nitrite production by RAW264.7 and U937 cells infected with *S.* Typhimurium wild-type strain ATCC 14028 s. RAW264.7 cells and U937 cells were infected at a MOI of 10, and nitrite levels in the supernatant were measured using Griess assays at the indicated time points ($n = 3$ independent experiments). **b** Bacteria burden of *S.* Typhimurium wild-type in RAW264.7 and U937 cells. Bacterial CFU ($\times 10^5$) /well (y axis) and time after addition of gentamicin (x axis) are indicated ($n = 3$ independent experiments). **c** Higher expression of *S.* Typhimurium SPI-2 genes in RAW264.7 cells than that in U937 cells. RAW264.7 cells and U937 cells were infected with *S.* Typhimurium wild-type for 8 h (MOI = 10), cell were then lysed and the intraceullar bacteria were collected for RNA extraction and RNA-seq (R8: RAW264.7 8 h; U8, U937 8 h). RNA extracted from bacteria in the RPMI-1640 medium was used as control (Con) ($n = 3$ independent experiments). The data are taken from Supplementary Table 1. All data are presented as mean ± SD. *P* values were determined using two-way ANOVA (**a, b**). hpi hours post-infection. Source data are included in Supplementary Data 1.

Differences in NO production and intracellular bacterial burden were further confirmed in primary mouse bone-marrow-derived macrophages (BMDMs) and in primary human peripheral blood mononuclear cells (PBMCs) (Supplementary Fig. 1c, d). The increase in bacterial burden was less in BMDMs (3.2-fold at 16 h) than in RAW264.7 cells (17.4-fold at 16 h), and this was likely due to the absence of inflammasome activation in RAW264.7 cells, which do not express the inflammasome adapter protein ASC.

As SPI-2 is required for *S.* Typhimurium replication in mouse macrophages[49], the induction of SPI-2 expression in RAW264.7 cells was confirmed by RNA-seq profiling (Fig. 1c and Supplementary Table 1) and quantitative real-time PCR (qRT-PCR) analysis of seven SPI-2 genes (*ssrA, ssrB, sipC, ssaE, sscA, sifA, ssaV*; Supplementary Fig. 1e). The expression of SPI-2 in U937 cells was also induced, but the level was significantly lower than that in RAW264.7 cells (Fig. 1c and Supplementary Table 1). Therefore, there is a positive correlation between host NO production and SPI-2 expression and replication levels of *S.* Typhimurium in mouse macrophages, implying that NO produced by the innate immune response might contribute to *S.* Typhimurium virulence during systemic infection in mice.

**Lack of host NO reduced *S.* Typhimurium SPI-2 gene expression and bacterial replication in mouse macrophages, and reduced bacterial burden in mouse systemic organs at early infection stages.** To investigate whether NO production in mouse macrophages contributes to *S.* Typhimurium virulence, RAW264.7 cells were treated with L-NMMA, a competitive inhibitor of iNOS, 2 h prior to infection to inhibit NO production, and the expression of SPI-2 genes in treated and non-treated cells was assessed at various time points during the infection period. Compared to untreated cells, the expression levels of *ssaG*, a representative SPI-2 gene, were significantly reduced in L-NMMA-treated RAW264.7 cells at each time point during the 24 h infection period as determined by qRT-PCR analysis (Fig. 2a). The same results were obtained using primary PMs from C57BL/6 mice (Fig. 2b). The addition of L-NMMA did not affect bacterial uptake by macrophages, as the treated and untreated RAW264.7 cells and PMs contained comparable number of intracellular bacteria 15 min post-infection (Supplementary Fig. 2a). Furthermore, the expression levels of *ssaG* were also significantly decreased in iNOS$^{-/-}$ PMs (iNOS-deficient macrophages from iNOS-immunodeficient C57BL/6 mice), that

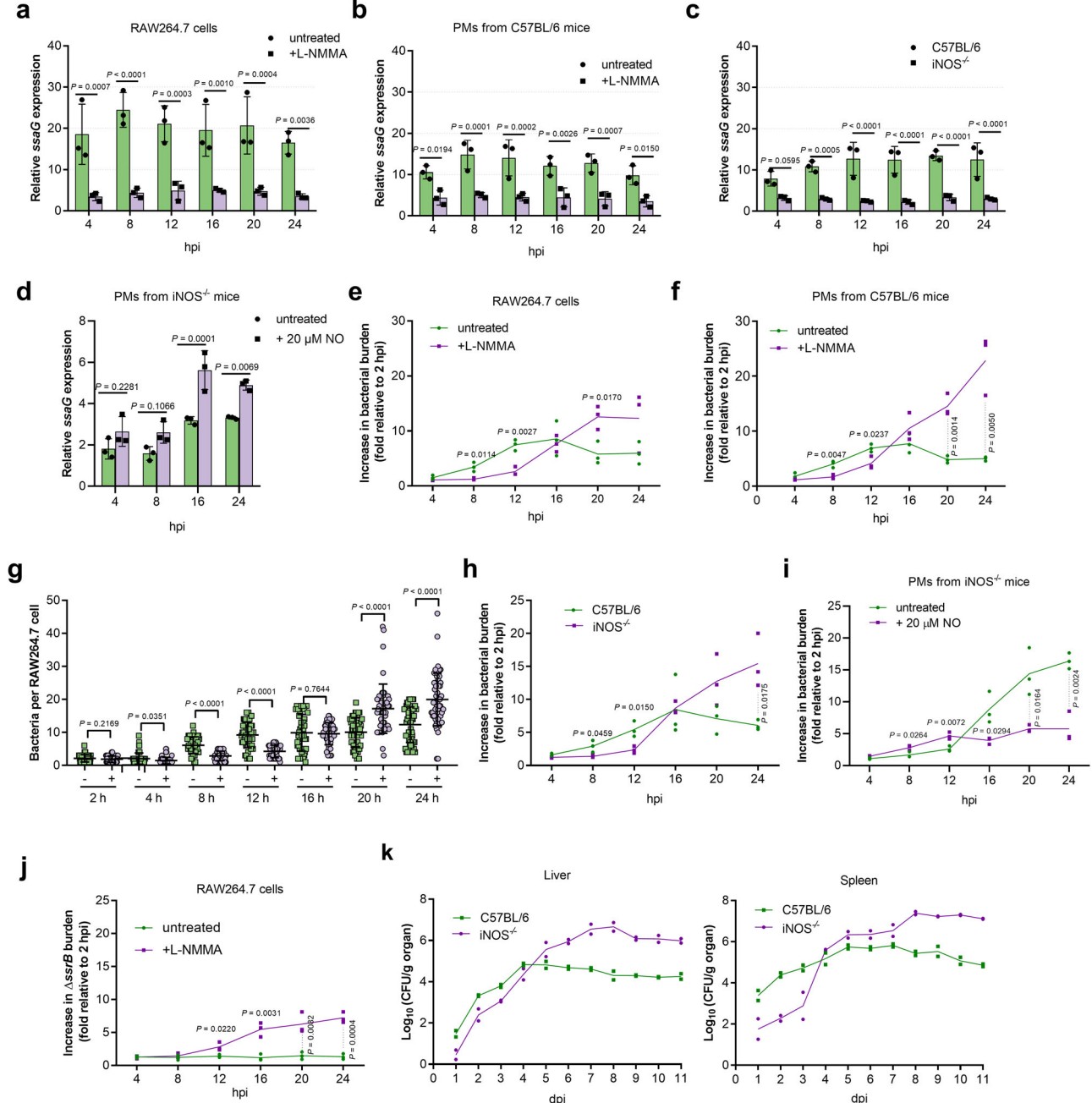

**Fig. 2 Lack of host NO reduced S. Typhimurium SPI-2 gene expression and bacterial replication in mouse macrophages, and reduced bacterial burden in mouse systemic organs at early infection stages. a–c** qRT-PCR analysis of *S*. Typhimurium *ssaG* mRNA levels in RAW264.7 cells (**a**), peritoneal macrophages (PMs) from C57BL/6 mice (**b**), and PMs from iNOS$^{-/-}$ mice (**c**) at the indicated time points in the presence or absence of 50 μM L-NMMA ($n = 3$ independent experiments). **d** qRT-PCR analysis of *ssaG* expression in PMs from iNOS$^{-/-}$ mice at 4, 8, 16, and 24 h post-infection (hpi) in the presence or absence of 20 μM NO ($n = 3$ independent experiments). **e, f, h** Increase in bacterial burden of *S*. Typhimurium wild-type in RAW264.7 cells (**e**), PMs from C57BL/6 mice (**f**), and PMs from iNOS$^{-/-}$ mice (**h**) at the indicated time points in the presence or absence of 50 μM L-NMMA ($n = 3$ independent experiments). **g** Number of intracellular bacteria per RAW264.7 cells ($n = 75$ cells per group pooled from three independent experiments). The number of bacteria per cell were counted in random fields. −, untreated; +, addition of 50 μM L-NMMA. Representative immunofluorescence images (8, 12, and 24 hpi) are provided in Supplementary Fig. 2b. **i** Increase in bacterial burden of *S*. Typhimurium wild-type in PMs from iNOS$^{-/-}$ mice at indicated time points in the presence or absence of 20 μM NO ($n = 3$ independent experiments). **j** Replication of *S*. Typhimurium *ssrB* mutant in RAW264.7 cells at the indicated time points in the presence or absence of 50 μM L-NMMA ($n = 3$ independent experiments). **k** Bacterial counts recovered from the liver and spleen of wild-type and iNOS$^{-/-}$ C57BL/6 mice intraperitoneally (i.p.) infected with *S*. Typhimurium wild-type at the indicated time points post-infection, $n = 2$ mice each day. All data are presented as mean ± SD. *P* values were determined using two-way ANOVA (**a–f, h–k**) or two-tailed unpaired Student's *t* test (**g**). hpi hours post-infection, dpi days post-infection. Source data are included in Supplementary Data 1.

are unable to produce NO, compared to that in wild-type C57BL/6 mouse PMs (Fig. 2c). The supplementation of NO increased $ssaG$ expression in iNOS$^{-/-}$ PMs (Fig. 2d). Thus, NO production in mouse macrophages promotes SPI-2 expression in *S.* Typhimurium.

As SPI-2 is required for *S.* Typhimurium replication in mouse macrophages, whether host NO production contributes to *S.* Typhimurium replication in mouse macrophages was investigated. Replication assays were carried out in untreated and L-NMMA-treated RAW264.7 cells and C57BL/6 mouse PMs. Compared to that in untreated cells, the intracellular burden of *S.* Typhimurium was significantly reduced in L-NMMA-treated cells at early infection stages (first 4 to 12 h) (Fig. 2e, f). Immunofluorescence enumeration further confirmed the presence of lower bacterial number in L-NMMA-treated RAW264.7 cells 12 h post-infection (Fig. 2g and Supplementary Fig. 2b). However, at later stages (20 and 24 h), when growth was inhibited in untreated cells, increased bacterial burden was detected in L-NMMA-treated cells (Fig. 2e–g and Supplementary Fig. 2b). Similar results were also observed in macrophages isolated from iNOS$^{-/-}$ mice (Fig. 2h). The supplementation of NO increased intracellular burden of *S.* Typhimurium in iNOS$^{-/-}$ PMs at early infection stages (first 4–12 h) (Fig. 2i). These results indicated that host NO production promotes *S.* Typhimurium replication in mouse macrophages at early infection stage, while inhibiting replication at later stage. To verify whether SPI-2 is required for the action of NO on *S.* Typhimurium replication, the $ssrB$ mutant, which is unable to express SPI-2 genes (Supplementary Fig. 2c), was generated and assessed for NO-dependent replication. Consistent with the essential role of SPI-2 in replication, the $ssrB$ mutant did not replicate in untreated cells (Fig. 2j). Interestingly, the $ssrB$ mutant did not replicate in L-NMMA-treated RAW264.7 cells at early infection stages but replicate at later stages (Fig. 2j), indicating that host NO production promotes *S.* Typhimurium replication at early infection stage by activating SPI-2 and inhibits replication at later stage in a manner that is independent of SPI-2 activation. Thus, NO is a host cue for SPI-2-dependent replication in mouse macrophages at early infection stages.

The contribution of host NO production to *S.* Typhimurium systemic infection was further investigated in vivo by comparing the bacterial burden in systemic organs (the liver and spleen) of i.p. infected C57BL/6 wild-type and iNOS$^{-/-}$ mice. ROS is reportedly essential for the early killing of ingested *Salmonella* by macrophages and RNS is involved in the control of *Salmonella* replication during the late stages of infection[30]. Consistently, both wild-type and iNOS$^{-/-}$ mice survived equally well for the first 6 days post-infection (Supplementary Fig. 2d). However, wild-type mice had higher bacterial burden in their liver and spleen than the iNOS$^{-/-}$ mice during the initial three (liver) or four days (spleen) post-infection (Fig. 2k), indicating that NO production promotes *S.* Typhimurium intracellular replication at the early infection stage, which was in agreement with the macrophage replication assays (in vitro results). In contrast, bacterial burden were significantly higher in iNOS$^{-/-}$ mice than in wild-type mice from day 5 post-infection (Fig. 2k), consistent with bacteriostatic activity of NO against *Salmonella* at later infection stages as reported previously[30]. The intravenous (i.v.) infected wild-type mice also had higher bacterial burden in their liver and spleen than iNOS$^{-/-}$ mice as determined at day 2 post-infection (Supplementary Fig. 2e), indicating that the NO-induced virulence promotion at the early infection stage is not associated with the infection route. These results confirmed that NO is an in vivo host signal that contributes to *S.* Typhimurium virulence by promoting intracellular replication at early infection stages.

**S. Typhimurium SPI-2 expression levels correlate positively with bacterial replication in mouse macrophages and in vivo virulence.** As host NO production demonstrated opposite effects on *S.* Typhimurium replication at early and late infection stages, the significance of NO contribution to *S.* Typhimurium replication at early infection stages was investigated. As NO promotes *S.* Typhimurium replication by inducing SPI-2 expression, the effect of SPI-2 expression levels on replication was assessed to determine the contribution of NO production. Seven $ssrA$ (a master regulator of SPI-2 genes) promoter replacement derivatives were generated (Supplementary Table 2). The promoters for the replacement of $ssrA$ promoter were selected based on their expression levels in RAW264.7 cells in the RNA-seq data. Their replacement of the $ssrA$ promoter were performed by overlap extension PCR and Red recombination system (see Materials and methods for details). These expressed SPI-2 genes in RAW264.7 cells at various levels ranging from 5.8-fold lower to 2.6-fold higher than the wild-type level ($P_{1150}$::$P_{ssrA}$, $P_{sinR}$::$P_{ssrA}$, $P_{yfiR}$::$P_{ssrA}$, $P_{aceB}$::$P_{ssrA}$, $P_{phnA}$::$P_{ssrA}$, $P_{2773}$::$P_{ssrA}$, and $P_{0658}$::$P_{ssrA}$) as verified by qRT-PCR analysis of $ssrA$ at 8 h post-infection (hpi) (Fig. 3a). The expression of four other representative SPI-2 genes ($ssaE$, $sscA$, $ssaG$, and $sifA$) inside RAW264.7 cells correlated well with the expression of $ssrA$ gene in each derivative strain (Supplementary Fig. 3a), indicating that the divergent expression of SPI-2 genes inside macrophages was achieved by using these derivative strains.

Macrophage replication assays were performed with the seven derivative strains in RAW264.7 cells and mouse PMs 16 hpi. At this time point, high intracellular bacterial burden and high NO production was detected (Fig. 1a and Supplementary Fig. 1c) while the death of cells was not significant (Supplementary Fig. 3b). We found that the increase in bacterial burden of wild-type *S.* Typhimurium and derivatives expressing similar ($P_{phnA}$::$P_{ssrA}$) or higher SPI-2 level ($P_{0658}$::$P_{ssrA}$ and $P_{2773}$::$P_{ssrA}$) were significantly higher than that of lower–SPI-2–expressing derivatives ($P_{1150}$::$P_{ssrA}$, $P_{sinR}$::$P_{ssrA}$, $P_{yfiR}$::$P_{ssrA}$, and $P_{aceB}$::$P_{ssrA}$) in both RAW264.7 cells and PMs, suggesting a positive correlation between SPI-2 expression levels and replication rates in macrophages (Fig. 3b, c). Lower–SPI-2–expressing derivatives ($P_{1150}$::$P_{ssrA}$, $P_{sinR}$::$P_{ssrA}$, $P_{yfiR}$::$P_{ssrA}$, and $P_{aceB}$::$P_{ssrA}$) grew as well as wild-type in LB medium (Supplementary Fig. 3c), and infected RAW264.7 cells as efficiently as wild-type (Supplementary Fig. 3d), indicating that the decreased replication ability of these derivatives in macrophages was not due to growth defect or a reduced infection ability. The results indicate that the replication level of *S.* Typhimurium in mouse macrophages is positively correlated with SPI-2 expression level.

Mouse infection assays were then carried out to further verify the effect of SPI-2 expression levels on *S.* Typhimurium virulence. The death rates of mice infected with wild-type *S.* Typhimurium and derivatives expressing similar ($P_{phnA}$::$P_{ssrA}$) or higher level of SPI-2 ($P_{0658}$::$P_{ssrA}$ and $P_{2773}$::$P_{ssrA}$) were significantly higher than that of those infected with lower–SPI-2–expressing derivatives ($P_{1150}$::$P_{ssrA}$, $P_{sinR}$::$P_{ssrA}$, $P_{yfiR}$::$P_{ssrA}$, and $P_{aceB}$::$P_{ssrA}$); all mice in the former group died by day 12 while 22–76% of mice in the latter group survived over the 20-day monitoring period (Fig. 3d). As expected, a negative correlation between SPI-2 expression levels and survival rates of infected mice was obtained (Fig. 3d). Consistently, SPI-2 expressing levels correlated positively with bacterial burden in the systemic organs (the liver and spleen) of infected mice on day 3 post-infection based on colony enumeration (Fig. 3e). The results indicate that the virulence of *S.* Typhimurium during systemic infection in mice is dependent on SPI-2 expression level. Thus, NO signaling is necessary for *S.* Typhimurium systemic infection by inducing high-level SPI-2 expression to enhance replication in mouse macrophages.

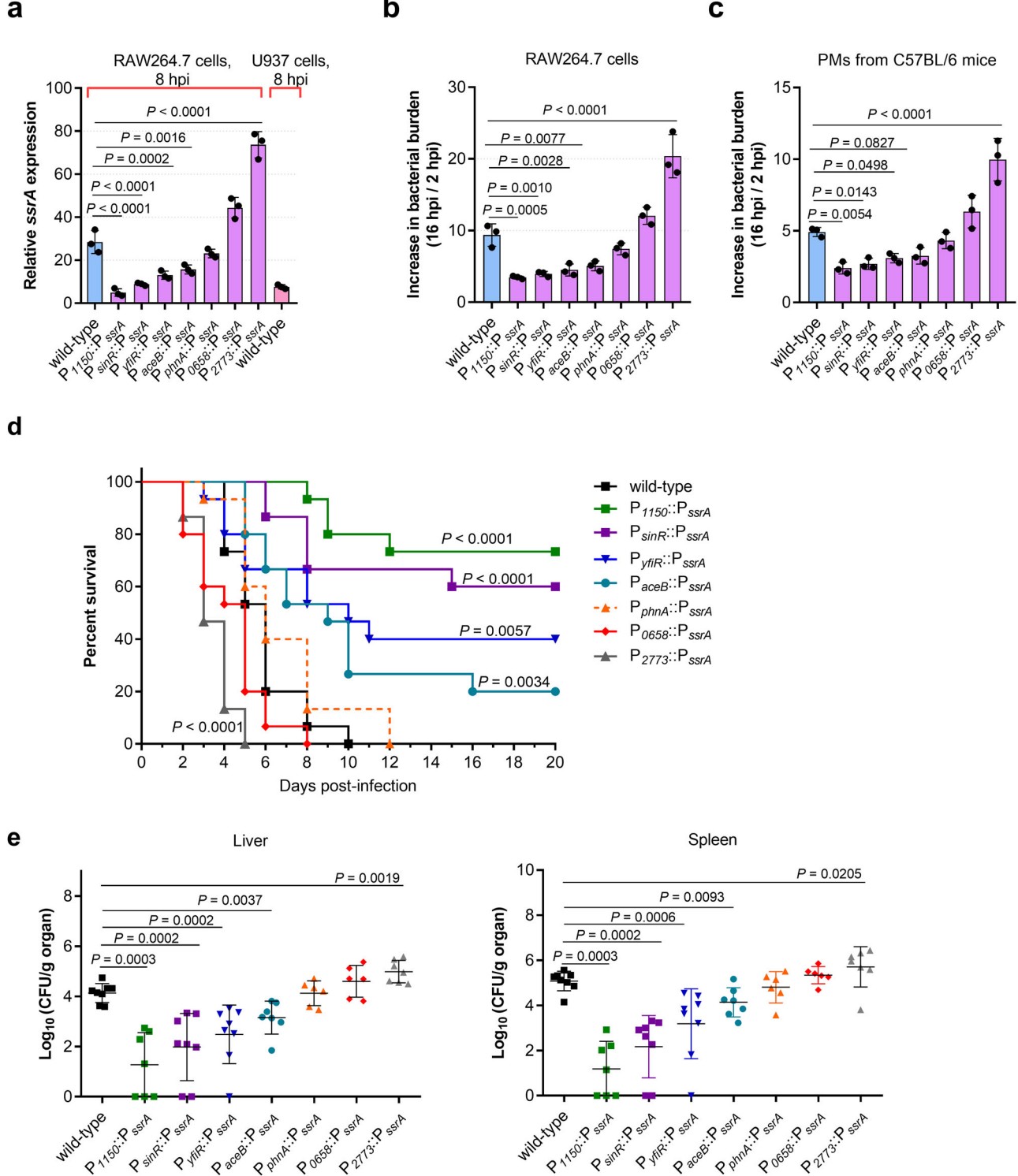

**Fig. 3 S. Typhimurium SPI-2 expression levels correlate positively with bacterial replication in mouse macrophages and in vivo virulence. a** qRT-PCR analysis of *S.* Typhimurium *ssaG* mRNA levels in different promoter–replaced strains (*n* = 3 independent experiments). RNA was extracted from bacteria collected from RAW264.7 cells or U937 cells at 8 hpi. RNA extracted from bacteria in the RPMI-1640 medium was used as a control. **b, c** Increase in bacterial burden of *S.* Typhimurium wild-type and promoter–replaced strains in RAW264.7 cells (**b**) and PMs derived from C57BL/6 mice (**c**) (*n* = 3 independent experiments). **d** Survival plots of mice after intraperitoneal (i.p.) inoculation with ~5×10³ CFUs of wild-type or promoter–replaced strains. *n* = 15 mice per group. **e** Bacterial counts recovered from the liver and spleen of mice i.p. infected with *S.* Typhimurium wild-type and promoter–replaced strains. *n* = 6–8 mice per group. All data are presented as mean ± SD. *P* values were determined using two-tailed unpaired one-way ANOVA (**a**–**c**), log-rank curve comparison test (**d**), or Mann–Whitney *U*-test (**e**). Source data are included in Supplementary Data 1.

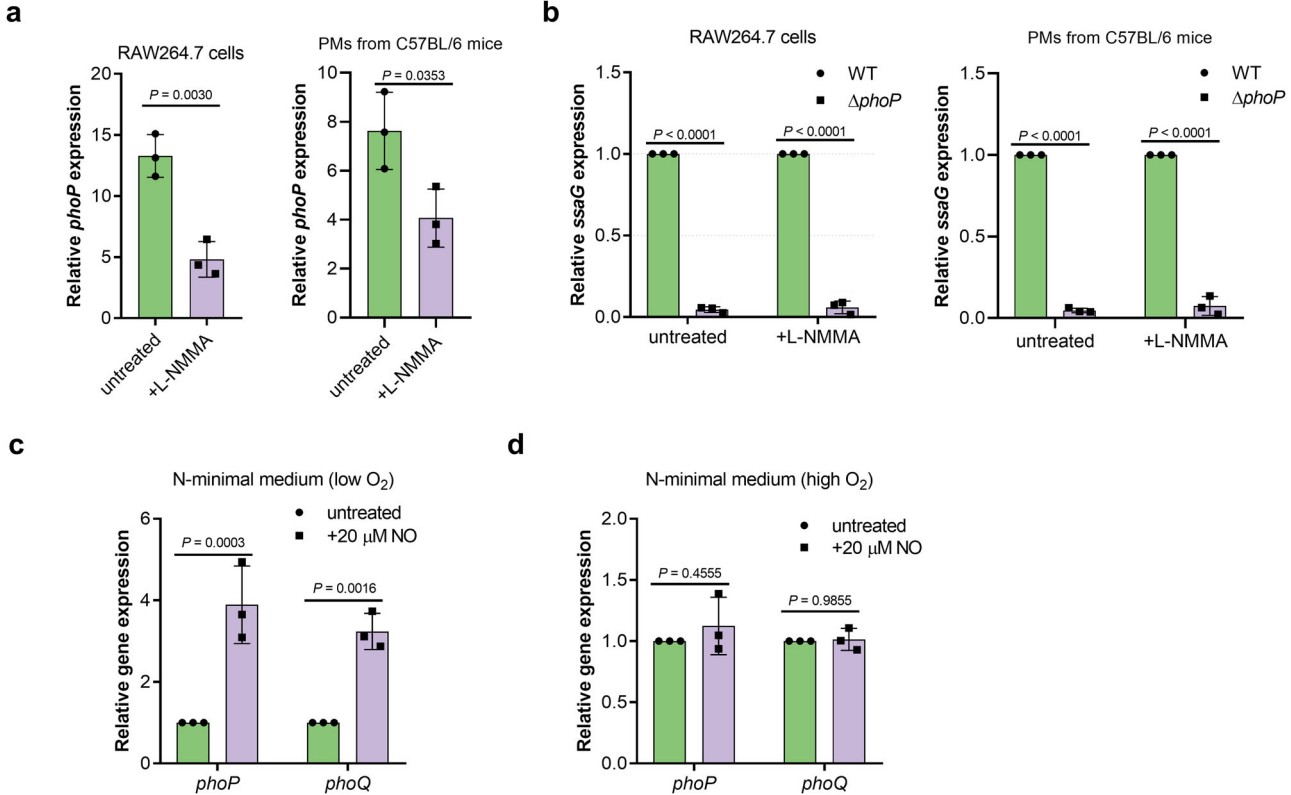

**Fig. 4 Host NO promotes SPI-2 expression by activating PhoP/PhoQ in mouse macrophages. a** qRT-PCR analysis of *S.* Typhimurium wild-type *phoP* mRNA levels in RAW264.7 cells or PMs in the presence or absence of 50 μM L-NMMA (*n* = 3 independent experiments). RAW264.7 cells or PMs were infected with wild-type strains at an MOI of 10. L-NMMA was added at a concentration of 50 μM. Fold changes in *phoP* gene expression in intracellular bacteria at 16 hpi relative to that in bacteria collected from the cell suspension are presented. **b** qRT-PCR analysis of *S.* Typhimurium *ssaG* mRNA levels in wild-type and *phoP* mutant in the presence or absence of 50 μM L-NMMA (*n* = 3 independent experiments). RAW264.7 cells or PMs were infected with wild-type or *phoP* mutant at an MOI of 10. L-NMMA was added at a concentration of 50 μM if indicated. At 16 hpi, fold changes in *ssaG* gene expression in wild-type relative to their expression in *phoP* mutant are presented. **c, d** qRT-PCR analysis of *S.* Typhimurium wild-type *phoP* and *phoQ* mRNA levels in the presence or absence of 20 μM NO either under low $O_2$ (**c**) or high $O_2$ conditions (**d**) (*n* = 3 independent experiments). Wild-type bacteria was grown in N-minimal medium in the presence or absence of 20 μM NO either under low $O_2$ or high $O_2$ conditions. Fold changes in *phoP* and *phoQ* expression in the presence of NO relative to that in the untreated samples are presented. All data are presented as mean ± SD. *P* values were determined using two-tailed unpaired Student's *t* test (**a**) or two-way ANOVA (**b**–**d**). Source data are included in Supplementary Data 1.

In the absence of NO signaling, SPI-2 expression was reduced 2.4-fold in RAW264.7 cells (Fig. 2a), which was similar to the SPI-2 expression level in the *ssrA*-replacement mutant P*yfiR*::P*ssrA* (Fig. 3a). It can be inferred that the attenuated virulence due to the lack of NO signal may be similar to that of P*yfiR*::P*ssrA*, which showed significantly reduced virulence in mice (Fig. 3d, e). NO production in RAW264.7 cells was not affected by the P*yfiR*::P*ssrA* mutation (Supplementary Fig. 3e). Thus, lack of NO production contributes, at least partially, to the reduced virulence in humans.

**Host NO promotes SPI-2 expression by activating PhoP/PhoQ in mouse macrophages**. To investigate the signal transduction pathway for NO-dependent SPI-2 activation, we first considered the possible involvement of known SPI-2 regulators. As indicated by RNA-seq and confirmed by qRT-PCR, the expression of *phoP* and *slyA* (a PhoP-regulated gene) was induced to higher level in mouse than that in human macrophages (Supplementary Table 3), in accordance with the differential induction of SPI-2 genes. Inhibiting NO production by L-NMMA in mouse RAW264.7 cells and PMs resulted in reduced *phoP* transcription (Fig. 4a), indicating that NO production enhanced *phoP* transcription in mouse macrophages. Considering the essential role of PhoP in SPI-2 induction in mouse macrophages, *ssaG* was not expressed by the *phoP* mutant in both untreated and

L-NMMA-treated RAW264.7 cells and PMs as expected (Fig. 4b), indicating that PhoP was required for NO-mediated SPI-2 activation. Therefore, higher SPI-2 expression in mouse macrophages was due to NO-enhanced PhoP activity. In vitro analysis further revealed that *phoP/Q* expression was significantly enhanced by NO (20 μM) under low $O_2$, but not high $O_2$, conditions (Fig. 4c, d), indicating that the response to NO under host condition (low $O_2$) by PhoP/PhoQ was likely mediated by other regulators, such as an anaerobic regulator.

**Fnr senses NO to enhance PhoP/Q-mediated SPI-2 activation in mouse macrophages**. The expression of *fnr*, which encodes a NO-responsive global anaerobic regulator, was induced to much higher levels in mouse RAW264.7 cells (14.6-fold) than that in human U937 cells (3.7-fold) based on the RNA-seq data (Supplementary Table 4), and inhibiting NO production in mouse RAW264.7 cells using L-NMMA resulted in markedly decreased *fnr* transcription levels (Fig. 5a), indicating the role of NO in *fnr* transcription. The expression of *phoP* and *ssaG* was down-regulated by the deletion of *fnr* and enhanced by its over-expression in RAW264.7 cells (Fig. 5b), indicating that Fnr is required for PhoP-induced SPI-2 activation in mouse macrophages. Down-regulation and up-regulation of *phoP* and *ssaG* expression by deletion and overexpression of *fnr*, respectively, were

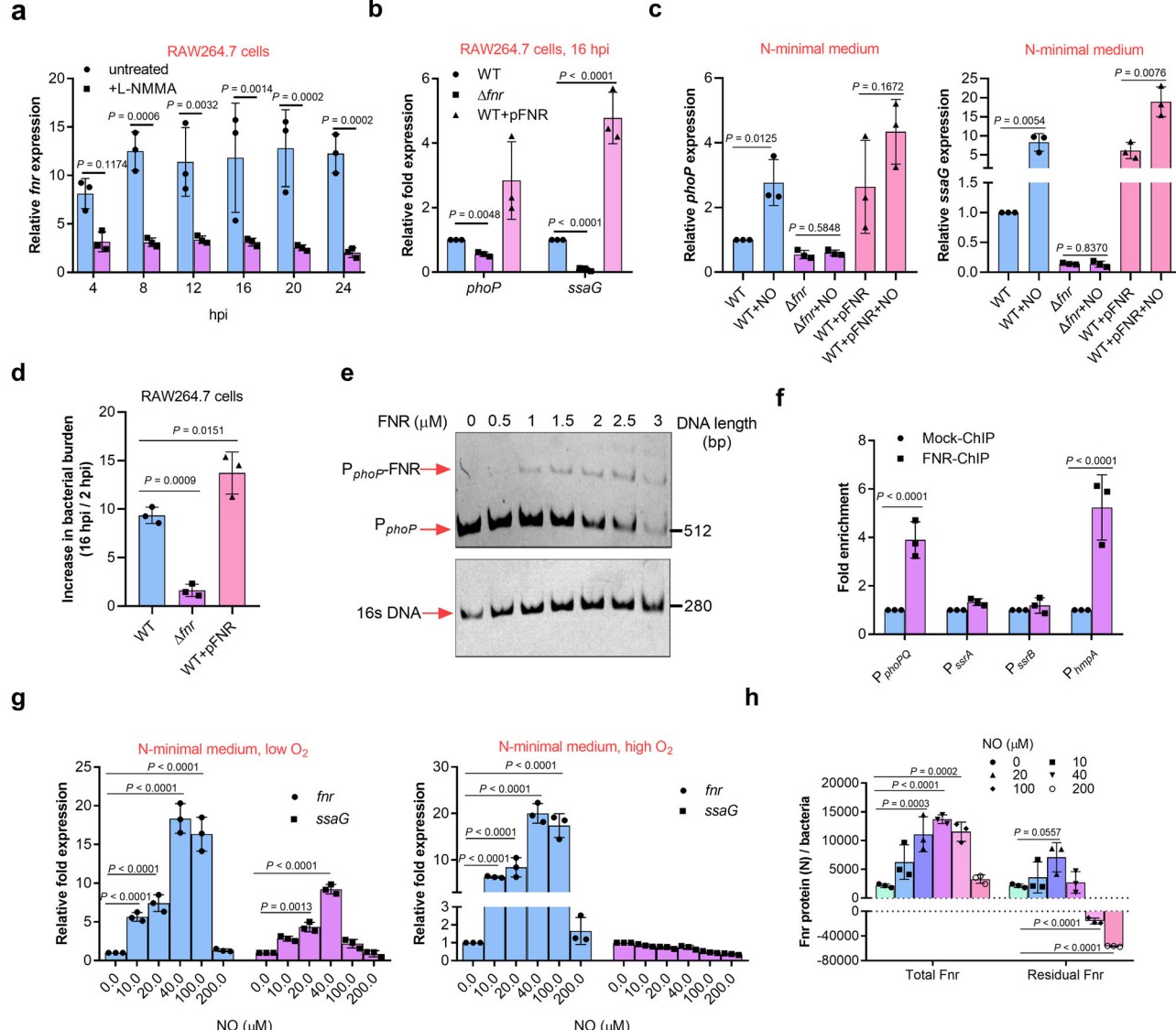

**Fig. 5 Fnr senses NO to enhance PhoP/Q-mediated SPI-2 activation in mouse macrophages. a** qRT-PCR analysis of *fnr* mRNA levels in *S.* Typhimurium wild-type at indicated time points post-infection of RAW264.7 cells, in the presence or absence of 50 µM L-NMMA ($n = 3$ independent experiments). **b** qRT-PCR analysis of *phoP* and *ssaG* mRNA levels in the wild-type, *fnr* mutant, or *fnr*-overexpressing strains at 16 h post-infection of RAW264.7 cells ($n = 3$ independent experiments). **c** qRT-PCR analysis of *phoP* and *ssaG* mRNA levels in the wild-type, *fnr* mutant, or *fnr*-overexpressing strains grown in N-minimal medium in the presence or absence of 20 µM NO ($n = 3$ independent experiments). **d** Increase in bacterial burden of wild-type, *fnr* mutant, or *fnr*-overexpressing strains in RAW264.7 cells at 16 hpi ($n = 3$ independent experiments). **e** Electrophoretic mobility shift assay (EMSA) of the *phoP* promoter with purified Fnr protein. Images are representative of three independent experiments. Full gels of the EMSA are shown in Supplementary Fig. 4. **f** Fold enrichment of the *phoP* promoter in Fnr-chromatin immunoprecipitation (ChIP) samples ($n = 3$ independent experiments). **g** qRT-PCR analysis of *fnr* and *ssaG* mRNA levels in the wild-type strain grown in N-minimal medium in the presence of indicated NO concentrations ($n = 3$ independent experiments). **h** Concentration of the Fnr protein in the presence of indicated NO concentrations ($n = 3$ independent experiments). *N* indicate the molecular number. Wild-type *S.* Typhimurium were grown in N-minimal medium in the presence or absence of the NO generator spermine NONOate and under low $O_2$ conditions. The total Fnr protein content in the bacteria was estimated using ELISA. Based on the reaction between [4Fe-4S] cluster of Fnr and eight NO molecules, the NO-inactivated Fnr molecular were calculated, and the residual functional Fnr protein molecular were calculated. All data are presented as mean ± SD. *P* values were determined using two-way ANOVA (**a–c**, **f–h**), or one-way ANOVA (**c**). Source data are included in Supplementary Data 1.

also detected in vitro in bacteria cultured in N-minimal medium under low $O_2$ condition in the absence of NO, and more strongly in the presence of 20 µM NO (Fig. 5c), indicating that Fnr-dependent PhoP/Q-mediated activation of SPI-2 is enhanced by NO production in mouse macrophages. *fnr* mutant showed significantly reduced bacterial burden in mouse macrophages, consistent with its role in activating *phoP* and SPI-2 genes (Fig. 5d).

Electrophoretic mobility shift assay (EMSA) analysis showed that purified Fnr protein bound to the promoter of *phoPQ* in vitro (Fig. 5e and Supplementary Fig. 4). Chromatin immunoprecipitation-quantitative PCR (ChIP-qPCR) analysis further demonstrated the binding of Fnr with the *phoPQ* promoter (Fig. 5f). These results indicate that Fnr directly activated *phoPQ* by binding to its promoter.

In vitro analysis further revealed that the expression of *fnr* increased with increase in NO concentration from 10 to 40 μM (derived from Spermine NONOate), corresponding to levels in mouse RAW264.7 cells (10–20 μM), but was not affected by higher level NO concentration (200 μM) (Fig. 5g). In addition to the NO concentration-dependent enhancement of *fnr* expression, *ssaG* expression was found to increase, but only under low $O_2$ condition, which is in agreement with the inactivation of Fnr under high $O_2$ conditions (Fig. 5g). Thus, Fnr senses NO production in mouse macrophages and enhances its transcription in response, leading to enhanced Fnr-mediated activation of SPI-2.

As binding to NO results in the inactivation of Fnr due to the nitrosylation of [4Fe-4S] cluster in the protein[44], the NO concentration-dependent enhancement of Fnr-mediated SPI-2 activation indicated the presence of unbound Fnr proteins under low NO concentration (10–40 μM) due to increased *fnr* transcription. This was confirmed by estimating the total Fnr molecules produced per bacterium[calculated from the protein concentrations determined by ELISA and molar mass of Fnr (Mr 30,000) and the portion needed for maximum NO-binding (each [4Fe-4S] cluster of Fnr reacts with maximum eight NO molecules)] (Fig. 5h).

## Discussion

In this study, we report that NO produced by the host innate immune system is an intracellular cue for the promotion of *S.* Typhimurium replication in mouse macrophages and mouse systemic organs, where it activates SPI-2 through Fnr- and PhoP/Q-mediated signal transduction pathway. The induction of NO signaling-induced SPI-2 activation in *S.* Typhimurium was concentration-dependent (5–20 μM), and SPI-2 expression was inhibited by high levels of NO (50–100 μM), indicating that *S.* Typhimurium virulence is differentially affected by the level of host NO production. The NO production in mouse macrophages was within the range required for maximum induction, while the lack of NO production possibly contributed to the absence of *S.* Typhimurium replication in human U937 cells, providing evidence for differential *S.* Typhimurium virulence in mice and humans.

In agreement with the finding in IFN-γ-activated macrophages, NO-dependent inhibition of *S.* Typhimurium replication at later infection stages was also detected in mouse macrophages in the present study, and further the inhibition was independent of SPI-2. Thus, host NO production acts as a double-edged sword, promoting *S.* Typhimurium virulence at early infection stages but inhibiting its replication at later stages. As NO is freely diffused into the SCV, it is likely that RNS derived from NO accumulate at high levels at later infection stages to inhibit *S.* Typhimurium replication. Supporting this hypothesis, the levels of $ONOO^-$ in RAW264.7 cells was significantly increased at 16 h post-infection (Supplementary Fig. 5), coincident to the onset of inhibition. However, further validation of this hypothesis and identification of the responsible congeners, if any, remain to be investigated.

Most of the known host environmental cues for SPI-2 activation in mouse macrophages, including mildly acidic pH, low $Mg^{2+}$, antimicrobial peptides, and hyperosmotic stress reported previously[50–52] as well as NO identified here, are mediated by PhoP/Q, further implicating PhoP/Q as an integrating point for multiple host cues for SPI-2 activation. Acidic pH, low $Mg^{2+}$, antimicrobial peptides, and hyperosmotic stress are directly sensed by PhoQ, and then, the activated PhoQ promotes the PhoP active (i.e., phosphorylated) state (i.e., PhoP-P). In contrast, *phoP/Q* response to NO is indirect and is mediated by Fnr that directly binds to the *phoPQ* promoter to enhance PhoP/Q expression. In addition, the NO-mediated activation of PhoPQ

occurs only under low $O_2$ conditions, which is also a characteristic of the SCV environment. Therefore, this study identified another mechanism for the control of PhoP/Q activity.

In addition to activate via SsrA/SsrB, PhoP also activates the SPI-2 effector SseL in an SsrA/SsrB-independent manner, via directly binding to the promoter region of *sseL* gene[53]. Besides PhoP, OmpR is another important regulator of SPI-2 genes[54]. In response to the increase of phosphorylated OmpR, the elevated SPI-2 *ssrA* expression in *S.* Typhimurium wild-type is comparable to that of the *phoP* mutant, suggesting that PhoPQ might also activate SPI-2 gene expression by modulating events upstream of OmpR[55]. Therefore, NO-mediated PhoP activation may also regulate SPI-2 independent of SsrA/SsrB.

The finding that low NO level could enhance Fnr-mediated SPI-2 activation was unexpected, as Fnr is inactivated by binding with NO[56]. Our previous understanding of the impact of NO on Fnr regulation was limited to de-repressing gene expression due to inactivation of the regulator, as in the case of Hmp[43,44]. The present study reports that NO not only inactivates Fnr by binding to the protein but also positively regulates its transcription at a lower concentration, resulting in enhanced Fnr activity and thereby affecting Fnr-regulated genes, including PhoPQ and SPI-2. Thus, NO signaling-induced Fnr activation has a global effect on genes regulated by this protein.

The finding that NO (at levels similar to those produced by host innate immune systems)-enhanced Fnr activation in a dose-dependent manner has a huge impact on Fnr-controlled genes expression, particularly in bacterial pathogens. In the case of *S.* Typhimurium, activation of Fnr by NO produced by mouse macrophages, therefore, also contributes to *S.* Typhimurium pathogenesis by affecting other genes in a global way. The impact of low level NO on Fnr activity is far beyond SPI-2 genes and has a global impact on the Fnr regulon.

Both Fnr and PhoP/Q are ancestral global regulatory systems that affect a large number of genes and are implicated in the pathogenicity of many bacterial pathogens apart from *Salmonella*, such as *Shigella*[57], uropathogenic *Escherichia coli*[58], and avian pathogenic *Escherichia coli*[59]. The involvement of these two systems in NO signal transduction pathway indicates that NO may also be exploited by other pathogens, provided that this molecules are produced by the respective hosts, which remain to be investigated.

Although NO production is negligible in human macrophages in most cases studied, there are a few exceptions. Previously published data suggests that human macrophages derived from healthy volunteers tend to lack detectable iNOS and produce negligible amounts of NO after stimulation by bacteria, whereas macrophages from diseased or stressed individuals have elevated levels of iNOS and NO[60]. Interestingly, there seems to be a correlation between NO production in human macrophages and *S.* Typhimurium systemic infection, as this pathogen causes systemic diseases in several immunocompromised individuals (very young, very old, or diseased) but not in healthy adults[61]. Invasive *S.* Typhimurium ST313 strains that emerged in sub-Saharan Africa also elicit bacteremia in the immunocompromised individuals in most reported cases[62]. Thus, systemic diseases caused by *S.* Typhimurium in immunocompromised humans may be partially associated with the production of NO.

Although SPI-2 is not required for the growth of human-specific serovar *S.* Typhi in human macrophages[63], it contributes to the immune evasion of *S.* Typhimurium in human macrophages, as that inactivation of SPI-2 in *S.* Typhimurium potentiates inflammasome responses of human macrophages, resulting in strong IL-1β production and macrophage death[64]. Although the supplementation of NO (20 μM) also increased *S.* Typhimurium SPI-2 expression in human macrophages

(Supplementary Fig. 6a), increasing SPI-2 expression (by using two higher–SPI-2–expressing derivative strains) in human macrophages resulted in only a small increase in intracellular burden of *S.* Typhimurium (Supplementary Fig. 6b, c). Therefore, the lack of NO-induced SPI-2 activation was only partially responsible for the low replication of *S.* Typhimurium in human macrophages, and other mechanisms are also involved in the control of its replication. Host cues and bacterial mechanisms for intracellular replication in human macrophages remain to be investigated.

The accumulation of large numbers of *S.* Typhimurium in mouse systemic loci (mainly liver and spleen), via bacterial replication in tissue macrophages, is required to cause lethal systemic disease, as that high abundance of bacteria would not be easily cleared by host immunity[65]. The massive increase of bacterial number is generally occurred at the initial infection phase (within a week), at which time the immune responses of liver and spleen are not well established[66,67], followed by a growth plateau phase occurred, as the host immunity begins to restrict bacterial growth[68]. The utilization of host NO production as a signal to promote intracellular replication during initial infection stage is likely an important mechanism for *S.* Typhimurium to induce systemic infection in mice.

*S. enetrica* serovar Dublin is capable of causing systemic infection in cattle, which is a major reservoir for *S. enetrica*, and also in mice[69,70]. Besides mouse macrophages, bovine macrophages were also reported to generate NO upon stimulation with bacterial constituents[27]. Whether the macrophage-derived NO contributes to the systemic infection of *S.* Dublin in the two hosts requires further investigation. *S.* Gallinarum and *S.* Choleraesuis can induce systemic infection, respectively, in poultry and pigs, which represent two other major reservoirs for *S. enetrica*[71]. However, both poultry and porcine macrophages stimulated by bacteria failed to produce detectable NO[27], indicating the absence of NO-signaling SPI-2 activation pathway. Therefore, other host-specific mechanisms may be utilized by *S. enetrica* to promote systemic infection.

In conclusion, this study demonstrated that NO produced by the host innate immune system acts as an important cue to promote *S.* Typhimurium virulence. This supports the hypothesis that host innate immune responses can be exploited by pathogens for their own benefit and that the interaction between the host innate immune system and pathogens are crucial for disease outcomes.

## Methods
**Ethics statement**. All animal experiments were approved by the Institutional Animal Care and Use Committee (IACUC) at Nankai University, Tianjin, China (IACUC number: 2018050601). We did our utmost to minimize the suffering of animals as well as the number of animals included in the study.

**Bacterial strains and plasmid construction**. The detailed information of the bacterial strains and plasmids used in this study is provided in Supplementary Table 5, and the primers used are listed in Supplementary Data 2. Mouse-virulent strain *S.* Typhimurium ATCC 14028s was used as the wild-type strain. Bacterial mutants were constructed by one-step gene replacement via red homologous recombination[72,73]. To generate mutants, the chloramphenicol or kanamycin resistance gene sequence and the two homologous arm sequences at both ends of target gene (38–40 bp) were PCR-amplified from pKD3 or pKD4 plasmid, respectively. To replace the *ssrA* promoter with other selected promoters, the promoter sequence of the selected gene was spliced with chloramphenicol resistance gene sequence by overlap extension PCR. The obtained DNA segments were then electroporated into the competent cells of wild-type *S.* Typhimurium carrying pKD46 plasmid that expresses a set of red recombinases. The mutants and promoter–replaced strains were screened on Luria–Bertani (LB) agar plates with the corresponding antibiotics. All established strains were confirmed by PCR amplification and DNA sequencing. A helper plasmid named pCP20 encoding the FLP recombinases was used to eliminate the antibiotic resistance gene, when required.

*fnr*-complemented strain and *fnr*-overexpressed strain were constructed by expressing *fnr* genes on the low-copy-number plasmid pWSK129. The DNA fragments bearing the *fnr* promoter region and open reading frame were PCR-amplified using high-fidelity Pfu DNA polymerase and wild-type *S.* Typhimurium genomic DNA as template. The purified DNA fragments and pWSK129 plasmid were double-digested with combined restriction enzymes (BamHI and EcoRI), and the enzyme-digested products were ligated using $T_4$ DNA ligase to generate the reconstructive plasmid pFnr. pFnr was then electrotransformed into *fnr* mutant strain or wild-type strain to generate the complemented strain and *fnr*-overexpressed strain, respectively.

**Bacterial growth conditions**. Bacterial strains were routinely grown in LB broth (10 g/L tryptone, 5 g/L yeast extract, and 10 g/L NaCl) at 37 °C except for strain bearing the temperature-sensitive plasmids pKD46 or pCP20, which were grown at 30 °C. To induce the expression of SPI-2 genes, bacteria were cultured overnight in LB broth and then diluted 1:100 using N-minimal medium (7.5 mM $(NH_4)_2SO_4$, 5 mM KCl, 0.5 mM $K_2SO_4$, 80 mM MES pH 5.8, 38 mM glycerol, 8 μM $MgCl_2$, 337 μM $KH_2PO_4$, and 0.1% casamino acids [w/v]). For non-inducing conditions of PhoP, the $MgCl_2$ concentration of N-minimal medium were changed to 1 mM. For low $O_2$ growth conditions, overnight–cultured bacteria were 1:100 sub-cultured into fresh LB medium or N-minimal medium (containing 8 μM $MgCl_2$ or 1 mM $MgCl_2$) in a 15 mL tightly closed polypropylene tube with no shaking for 6 h to reach the stationary phase ($OD_{600} ≈ 0.4$). For high $O_2$ growth conditions, overnight–cultured bacteria were 1:100 sub–cultured into fresh LB medium or N-minimal medium (8 μM $MgCl_2$ or 1 mM $MgCl_2$) with shaking at 200 rpm for 4 h to reach the stationary phase ($OD_{600} ≈ 0.8$).

When necessary, individual or a combination of antibiotics were used at the following working concentrations: 20 μg/mL chloramphenicol (Cm), 100 μg/mL ampicillin (Ap), 50 μg/mL kanamycin (Km), 200 μg/mL streptomycin (Sm), and 10 or 100 μg/mL gentamicin (Gm).

**Salmonella infection of mice**. Wild-type C57BL/6 mice were purchased from Beijing Vital River Laboratory Animal Technology Co. Ltd (Beijing, China). Congenic iNOS$^{-/-}$ mice with a C57BL/6 background were purchased from the Jackson Laboratory (USA). All mice were raised in a specific sterile environment in compliance with the scientific guidelines for animal research.

For mouse survival assays, overnight–cultured wild-type and *ssrA* promoter–replaced strains were sub–cultured into fresh LB medium with an inoculation ratio of 1:100. When the $OD_{600}$ value of the media reached 2, the bacteria were collected and serially diluted in 0.9% NaCl to $1 × 10^5$ CFU/mL or $2.5 × 10^4$ CFU/mL. Groups (5–10 mice/group) of 6–8 weeks–old C57BL/6 mice were infected with the indicated *S.* Typhimurium strains by injecting 0.1 mL of the 0.9% NaCl suspension containing $1 × 10^4$. The mortality of the infected mice was recorded at the same time every day, and the daily survival rate was calculated.

To calculate the number of *Salmonella* colonized in the liver and spleen, the infected mice were sacrificed by cervical dislocation three days post-infection. The liver and spleen were excised using sterile scissors and tweezers, and the respective tissues were ground in PBS using a homogenizer. The homogenates were diluted and then spread on LB agar plates to count the CFU.

**Macrophage culture**. The mouse macrophage cell line RAW264.7 (ATCC TIB71), human macrophage cell line U937 (ATCC CRL-1593.2) and THP-1 (ATCC TIB-22) were purchased from the Shanghai Institute of Biochemistry and Cell Biology of the Chinese Academy of Sciences (Shanghai, China). Primary human peripheral blood CD14 + mononuclear cells (PBMCs) were purchased from Stem Cell Technologies (Vancouver, BC, Canada; https://www.stemcell.com/human-peripheral-blood-cd14-monocytes-frozen.html). Macrophages were cultured in RPMI-1640 medium (Gibco) containing heat–inactivated fetal bovine serum (FBS; Gibco) at 37 °C under a 5% $CO_2$ atmosphere. Approximately 48 h before infection, RAW264.7 cells were suspended and seeded into 12-well cell culture plates, with or without coverslips, at a density of $1 × 10^5$ cells per well. The U937 cells and THP-1 (ATCC TIB-22) were activated with phorbol 12-myristate 13-acetate (Sigma–Aldrich) at a concentration of 10 nM for 48 h before being infected[48]. This treatment allowed the cells to become adherent and activated. The CD14 + mononuclear cells were differentiated to macrophages by the administration of recombinant 1% macrophage colony–stimulating factor for 7 days before infection.

Mouse peritoneal macrophages (PMs) were elicited by injecting sodium periodate solution into the peritoneal cavity of C57BL/6 mice and/or congenic iNOS$^{-/-}$ mice and then harvested. The harvested PMs were incubated in RPMI-1640 medium containing 10% FBS for 3 h at 37 °C under 5% $CO_2$ environment. Then, the cells were washed twice with PBS and the adherent cells were further cultured in RPMI-1640 medium containing 10% FBS for 48 h. Primary bone-marrow-derived macrophages (BMDMs) were isolated from by flushing the femur and tibia with PBS. The harvested BMDMs were incubated in RPMI-1640 medium containing 10% FBS and 100 ng/mL M-CSF. Cells were incubated for 8 days at 37 °C under 5% $CO_2$ environment with medium change every 2–3 days. All cells were cultured in antibiotics–free media.

**Assessment of intramacrophage growth of *Salmonella*.** The macrophage replication assays were performed using the stationary phase of *Salmonella*[74]. Overnight–cultured bacteria were inoculated into fresh LB medium with a ratio of 1:100 and further cultured till the $OD_{600}$ value reached approximately 2.0. The bacterial culture was centrifuged at 5000 rpm for 5 min, and the supernatant was discarded while the bacterial pellet was resuspended in 10% normal mouse serum and incubated at 37 °C for 20 min of opsonization. Then, the bacteria were added to the cell plates with macrophage monolayers at a multiplicity of infection (MOI) of 10. The cell plates were centrifuged at $1000 \times g$ for 5 min to facilitate the invasion process. After 30 min of incubation at 37 °C under 5% $CO_2$, the cell culture supernatant was discarded and the infected cells were gently washed three times with PBS (warmed to 37 °C). Then, the cells were cultured in a medium containing 100 µg/mL gentamicin for 1 h at 37 °C, followed by culturing the cells in a medium containing 10 µg/mL gentamicin till the end of the infection process. At the indicated time points post-infection, the culture supernatants in the cell plates was removed and the cells were gently washed three times with PBS. Finally, 0.1% Triton X-100 was added to the cell plates and the cells were lysed by vigorous aspiration. The lysates was serially diluted and plated onto LB agar plates to count the CFU number of *Salmonella*. At each time point post-infection, the number of viable cells/well was detected. The bacterial CFUs were normalized by the numbers of viable cells. The increase in bacterial burden of *Salmonella* was calculated as the number of intracellular bacteria at the indicated time points divided by that at 2 h. To investigate the effect of NO on *Salmonella* growth in cells, 50 µM L-NMMA (l-monomethylarginine) was added 2 h prior to infection to inhibit the possible induction of SPI-2 expression.

**Determination of NO production.** As NO is an end product of nitrite, its concentration was indirectly measured by detecting that of nitrite using the Griess reagent. At the indicated time points post-infection, the culture supernatants (50 µL) of RAW264.7 and U937 cells were transferred into 96 well plates, followed by the addition of 100 µL Griess reagent and incubation at room temperature for 10 min. Then, the absorbance at 570 nm was measured using the Spark multimode microplate reader (Tecan) and the nitrite concentration in the culture supernatant was calculated based on the sodium nitrite standard curve.

**Effect of NO on gene expression of *Salmonella* in vitro.** In vitro gene expression of *ssaG*, *fnr*, and *phoP* was assessed using qRT-PCR in bacteria cultured in N-minimal medium with low $Mg^{2+}$ concentration (8 µM $MgCl_2$) and/or SPI-2 non-inducing (1 mM $MgCl_2$) N-minimal medium as described above. Gene expression was also assessed in bacteria cultured in N-minimal medium containing 8 µM or 1 mM $MgCl_2$ under either low or high $O_2$ conditions. Spermine NONOate (Abcam) was used as the NO donor, and the effect of NO on target gene transcription was investigated. The NO donor was added to the culture after 1 h of bacterial growth in 8 µM or 1 mM $MgCl_2$ N-minimal medium, when the bacterial cells were in the late log phase.

**qRT-PCR analysis.** qRT-PCR was performed in a QuantStudio 5 Real-Time PCR system (Applied Biosystems). RNA was reverse transcribed to cDNA using SuperScript II (Invitrogen) with random hexamers (Sigma) according to the manufacturer's instructions. The qRT-PCR reaction system was carried out in a total volume of 20 µL containing 1 µL cDNA, 200 nM of each primer, and 10 µL of universal SYBR Green Master mix. The qRT-PCR reaction procedure was set to 95 °C for 10 min, followed by 40 cycles of 95 °C for 15 s and 60 °C for 30 s. The fold change of target gene expression relative to that of the housekeeping gene (*16S rRNA*) was determined by the $2^{-\Delta\Delta Ct}$ method.

**RNA-Seq and gene expression analysis.** To study and analyze the gene expression of *Salmonella* inside different types of macrophages, RNA was extracted from the intracellular bacteria at 8 h post-infection of RAW264.7 and U937 cells. RNA extracted from bacteria in the RPMI-1640 medium was used as control. RNeasy Mini Kit (Qiagen) was used for RNA purification, which included an on-column DNase digestion process. Host cells RNA was depleted using MicrobEnrich kit (Ambion), and then bacterial 23S and 16S rRNA was depleted using MicrobExpress kit (Ambion). Capillary electrophoresis (Agilent) was used to determine the RNA quality, and the RNA was quantified using a NanoDrop 2000 (NanoDrop Technologies). It was then reverse transcribed to cDNA using SuperScript Double–Stranded cDNA Synthesis Kit (Invitrogen).

cDNA libraries for Illumina platform sequencing were constructed using the mRNA-Seq 8-Sample Prep Kit (Illumina) according to the instructions provided, and sequencing was performed on the Illumina HiSeq 2000 platform by Novogene Co., Ltd. (Beijing, China). A final dilution of 10 pM of the cDNA library was loaded onto the sequencing machine. Phred quality scores less than 20 were regarded as low-quality reads and discarded, which were assessed using the FastQC quality control tool with default parameters. The treated reads were then mapped to the genome of *S.* Typhimurium ATCC 14028S (CP001363 and CP001362) Using Bowtie (version 2.2.3). The number of reads mapped to each gene was obtained using HTSeq (version 0.6.1). The gene expression levels were calculated as reads per kilobase of transcript per million mapped reads (FPKM). Differentially expressed genes were determined using R packages DESeq2 with FDR significance thresholds set as *P* value <0.05 and fold-change >2.

**Immunofluorescence staining.** RAW264.7 cells were seeded on glass coverslips and infected as described above. At indicated time points post-infection, the infected cells were fixed with 3% paraformaldehyde (PFA) for 15 min and washed three times with PBS. Cells were then permeabilized for 20 min in 0.1% Triton X-100 solution and washed with PBS, followed by blocking with 5% BSA in PBS for 30 min. The cells were then incubated with mouse anti-*Salmonella* LPS (Abcam) antibody that was diluted 100 times with PBS for 1 h. After washing the cells three times with PBS, they were incubated with goat anti-mouse IgG (FITC) (Abcam) secondary antibody, diluted 200 times with PBS, for 1 h. After repeating the wash process, the cells were incubated with DAPI (Invitrogen) for 2 min. After a final washing process, cells were covered with mounting medium. A confocal laser scanning microscope (Zeiss LSM800) were used for the observation of intracellular bacteria and image acquisition. The ZEN 2.3 (blue edition) were used for the further image processing.

**Cytotoxicity assays.** RAW264.7 macrophages were seeded in 96-well cell culture plates and infected with *Salmonella* as described above. At indicate time points post-infection, the supernatant was aspirated and used to determine the concentration of the released cytoplasmic lactate dehydrogenase (LDH) using CytoTox 96® Non-Radioactive Cytotoxicity Assay (Promega). Cytotoxicity was presented as the percentage of LDH released relative to the maximum release from lysed macrophages.

**Expression and purification of Fnr-His₆.** The *fnr* sequence was cloned into the pET-28a plasmid to obtain the pET-Fnr plasmid, which was elctrotransformed into *Escherichia coli* BL21 to express Fnr fusion proteins. Fnr fusion proteins were purified from the soluble extracts of the pET-Fnr plasmid–expressing BL21 using a HiTrap $Ni^{2+}$–chelating column. Bradford method was used to determine the protein concentration. The final purified protein was aliquoted and stored at −80 °C.

**Detection of the Fnr protein from bacterial cell extracts by ELISA.** The Fnr protein polyclonal antibody was produced by Wuhan bioyears Biotech Co., Ltd (Wuhan, China). Overnight *S.* Typhimurium wild-type strain were 1:100 sub-cultured in N-minimal medium and grown under low oxygen conditions without and with different concentrations of Spermine NONOate (5, 10, 20, 50 and 100 µM) to an $OD_{600}$ of 0.6 (~$5 \times 10^8$ bacteria/mL). The bacteria were harvested and the pellet was suspended in the Laemmli sample buffer[75] containing 4% SDS. The suspension (about 40 g protein/L) was sonicated for 20 s. The cell extracts (0.2 mg protein) were subjected to SDS gel electrophoresis and were then transferred to polyvinylidene difluoride (PVDF) membranes. Fnr protein was detected by ELISA using anti-Fnr (diluted 1:250 to 1:500) and a peroxidase coupled secondary antibody (Abcam). The amount of Fnr protein in cell extracts was estimated from the stain produced in ELISA after separation by SDS gel electrophoresis. The estimation was done by comparing the intensity of the stain produced by the unknown sample and by known amounts (0.05–0.5 µg) of isolated Fnr protein in the same gel.

**EMSA.** PCR fragments containing the promoter region sequence of *ssrA*, *ssrB*, *phoP*, and *hmpA* were amplified by PCR using genomic DNA of *S.* Typhimurium 14028 S as template. EMSAs were performed by incubating the purified promoter fragments (40 ng) with increasing concentrations of purified Fnr-His₆ protein (0–20 µM) at 37 °C for 20 min under low $O_2$ environment. The total reaction mixture was 20 µL, and the binding buffer was comprised 20 mM Tris-HCl (pH 7.5), 80 mM NaCl, 0.1 mM EDTA, 1 mM DTT, and 5% glycerol. Samples were loaded with the native binding buffer on a 6% polyacrylamide native gel in 0.5× Tris-borate-EDTA. Ethidium bromide was used for staining the DNA fragments.

**ChIP and ChIP-qPCR.** Overnight cultures of FLAG-tagged bacterial strains were diluted 1:100 into N-minimal medium and continued to grow to an $OD_{600}$ of approximately 0.6. Bacterial cultures were treated with 1% formaldehyde for 25 min at room temperature. The reaction was quenched by the addition of glycine at a final concentration of 0.5 M, and the samples were pelleted by centrifugation at 12,000 rpm, 4 °C for 2 min. Next, the bacterial pellets were washed three times with ice-precooled PBS. The samples were then used for ChIP in accordance with the guidelines of the Chromatin Immunoprecipitation kit (Millipore). A mouse monoclonal antibody against FLAG (Sigma; 1:1000 dilution) was used. Untreated chromatin was de-cross-linked by boiling for 10 min and purified as the "input" control for ChIP-qPCR assays. The value of the immunoprecipitated DNA, quantified by qPCR, divided by that of the input unprecipitated DNA was considered the relative enrichment of candidate gene promoters. To account for non-specific enrichment, these values were normalized to the values obtained for each promoter using untagged wild-type.

**Statistics and reproducibility**. The in vitro experiments were performed in duplicate and repeated at least three times ($n \geq 3$), unless specifically stated. Mouse virulence assays were conducted twice with at least 2 mouse ($n \geq 2$) in each injection group, and the combined data for the two experiments was used for statistical analysis. Statistical significance was analyzed with GraphPad Prism 8.0.1 software (GraphPad Inc., San Diego, CA) using the two-tailed unpaired Student's $t$ test, one-way ANOVA, two-way ANOVA, log-rank (Mantel–Cox) test, or Mann–Whitney $U$-test according to the test requirements, as stated in the figure legends.

**Reporting summary**. Further information on research design is available in the Nature Portfolio Reporting Summary linked to this article.

## Data availability

The RNA-seq acquired in this study are available in the NCBI Sequence Read Archive (SRA, PRJNA915482). Source data are provided in Supplementary Data 1.

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

## Acknowledgements
This work was funded by the National Natural Science Foundation of China (32130003, 81871624, 32170110, 32070133), the National Key R&D Program of China (2018YFA0901000), the Natural Science Foundation of Tianjin (22JCYBJC01060), and the Natural Science Foundation of Shenzhen (JCYJ20210324135007019).

## Author contributions
L.F. conceived and supervised the research; L.J., W.L., X.H., S.M., X.M., X.Y., and B.Y. performed the research; D.H. and B.L. provided technical support and insights; L.J. analyzed the data; and L.J. and L.F. wrote the manuscript.

## Competing interests
The authors declare no competing interests.
