## [Peer Review File · Communications Biology]

Reviewers' comments:

Reviewer #1 (Remarks to the Author):

Here, Jiang et al present a comprehensive study to show that in mice, NO promotes replication of *S. Typhimurium* in the early stages of infection (i.e. up to day 5 post-infection in vivo and 12 hours post-infection in vitro) by activating the SPI-2 type III secretion system, a process known to facilitate intracellular survival of *Salmonella enterica* serovars. The authors further show that SPI-2 activation is mediated by Fnr and the PhoP/Q two-component regulatory system. This work would be of interest to the general *Salmonella* field as it is dependent on SPI-2 machinery. It may be of interest to those working on other intracellular bacterial pathogens that utilise a type 3 or 4 secretion system (e.g. *Shigella*, *Legionella* or *Coxiella*). It would be of specific interest to those in the *Salmonella* field who use both in vitro and in vivo models of infection to confirm mechanisms of virulence. The work is novel, but is limited to the mouse model of infection and outcomes relating to this. The work has been executed well, the statistical methods applied are sound, mouse numbers are sufficient in most figures. I have comments that would need to be addressed before the article could be accepted for publication, but overall, this is a valuable study for the field of bacterial pathogenesis and adds insight into bacterial mechanisms of disease and also host resistance to infection.

Major comments:

1. In the abstract: The authors overstate the application of the study a little (new pathway, important, etc, but it's only in mice) – these novel observations in mice should be published because they are not relevant to the human disease model, and so should be taken into consideration when researchers are making conclusions and assumptions about disease outcomes for humans when using the mouse model of *Salmonella* infection.
2. In the introduction: I am not a great believer in the idea that mouse infection with STm is a robust model for human Typhi infection, as each bacterium has a significantly different genetic makeup. This paper should probably make the argument that it could draw parallels in host responses during human Typhi infection and mouse STm infection so that it MAY be relied upon as a model for Typhi or maybe better still, invasive STm serovars. This is currently not clear in the introduction.
3. Do the authors see similar dynamics of NO production in both RAW264.7 and U937 with heat-killed STm and /or LPS treatment? This is another way of testing the involvement of the SPI-2 T3SS components.
4. The cell lines compared (RAW264.7 vs U937) have some very innate differences that can influence the replication dynamics. RAW264.7 cells are missing components of the inflammasome, a critical innate detection and response mechanism against intracellular pathogens. Do the authors have any data from mouse bone-marrow-derived macrophages to support the replication dynamics they observe in RAW264.7 cells over the chosen infection time course – this could be placed as a supplemental figure. How can the authors determine that the lack of control of STm is not due to the lack of inflammasome activation (which would be occurring in U937 cells).
5. In Fig 2D, there is no explanation for why 16 hrs post-infection was selected, was this just as a mid-infection test or did other time points show similar results?
6. Line 209: why ip infection rather than physiological oral infection? You will still see dissemination to peripheral organs via oral infection route so not sure why only ip was used?
7. Does the PyfiR::PssrA strain induce less nitrite in vitro? (as per assay in Fig 1A?) This would help support their conclusions on lines 257-260.
8. Fig 4A does not indicate at what time point this analysis was performed? Fig 4B indicates analyses were performed at 16 hpi, was this the same for 4A? And what about PMs from C57BL/6 mice? Do the authors see similar results in PMs?
9. The authors used 2 mouse macrophage cell lines to show results, but only one human macrophage cell line. Do the authors have any data in THP-1 macrophages or human PBMCs to reduce the likelihood of a cell specific effect? I don't expect to see all results repeated in another cell line, but one significant result showing lack of NO production in another human cell line would be appreciated and increase the robustness of the results.

10. Can the authors comment on whether the NO pathway is active in other species that are major reservoirs for *Salmonella enterica*, e.g. pigs, cows? This may add some weight to the relevance of the findings.

Minor comments:

1. Figure 1C: legend has no details as to the type RNA-seq undertaken at what time point of infection the RNA was processed. Supp Fig 1 should be amended accordingly.
2. Fig 2A, B and C: keep y axis the same, i.e. all max at 30 – it is more transparent, otherwise the reader could assume similar levels of overall expression at first glance. Same for Fig 2E and F, maybe even Fig 2H and I also.
3. The strain name of the STm being used should be mentioned in both the methods and in the results (just once at the start – not all the way through) or at least in the figure legends.
4. In Fig 2J, it is not immediately clear that this is an *ssaB* mutant being used. I suggest including a reference to it in the figure (i.e. on the y axis legend, say fold intracellular replication of Δ *ssaB*).
5. In Fig 2G – no details for what stain was used to count bacteria. Regardless of whether these details are in the methods or not – there should be some indication of how this was done withing the figure or the results text so that the reader can immediately decide whether this was a reliable/viable/robust assessment.

Line 43: add 'the' in front of 'mouse model of *S. Typhimurium*...'

Line 46: change the word 'aspect' to 'hand'

Line 56: Add 'The' before 'SPI-2-encoded...'

Line 57: change to 'outside of the SPI-2 operon, ...'

Line 58: add 'the' before 'Salmonella-containing vacuole...'

Line 66: add 'observed' before 'in mouse macrophages'

Lines 85 and 86: add 'the' before 'transcriptional level' and before 'posttranscriptional level'.

Line 104: change start of sentence to "In this study, the contribution of host NO production to *S. Typhimurium* infection in mice was investigated using ..."

Line 115: add 'the' before 'innate immune...'

Line 141: change 'modest' to 'modestly'

Line 158: Fig 2D reference is in bold while others are not.

Line 222: change 'indicated' to 'indicate'

Line 241: add 'the' after the word 'that'

Line 305: change 'a' to 'an'

Line 305: change 'level' to 'levels'

Line 315: change 'consisting' to 'consistent'

Line 357: add the word 'the' before 'host innate immune'

Line 361: change 'level' to 'levels'

Line 365: remove the word 'an'

Line 370: add 'the' before 'SCV'

Line 389: change 'report' to 'reports'

Line 405: remove the word 'of'

Line 411: remove the word 'the'

Line 423: add 'the' before 'host innate immune'

Line 468: change 'mouse' to 'mice'

Reviewer #2 (Remarks to the Author):

In their manuscript, Jiang and colleagues explore the role of nitric oxide as an innate immune cue during *Salmonella* infection. They suggest that production of nitric oxide in murine macrophages (but not in human macrophages) leads to Spi2 induction and promote *Salmonella* replication at early time-point. Their results also show that this is the opposite at later time points, and that NO is required to

control infections. Their data suggest that PhoP and Fnr regulate Spi2-related genes expression in response to NO.

Overall the observations presented in the paper are interesting and will be of interest to the field. They provide interesting genetic approaches to validate their hypothesis. However, I feel there is some caveats that will need to be fixed before acceptance of this publication. I hope my comments will be useful to the authors.

Major points:

Figure 1: The claim that mouse but not human induce NO during Salmonella infection is based on the usage of cell lines. Although these are good resources, conclusions cannot be drawn as they do not behave in a similar manner as primary cells. For example, RAW cells do not express competent inflammasome (lack ASC/PYCARD) whereas U937 express competent inflammasome component. This should be tested in primary macrophages. It has been shown by some studies (PMID: 23774601) that primary human macrophages can produce NO. The authors should be careful about drawing strong conclusion from cell lines.

The authors used a bunch of Salmonella mutants (various promoters to control Spi2 expression). Their approach is certainly interesting but does not fully imply that Spi2 is the main element responsible here. Why not simply include an *ssaR* ko strain. This strain is commonly used as a Spi2 inactive mutants and would be an important control here. This would be a good control for their infection model (although I would understand this may represent too much work).

Figure 3B/C: Why looking at the replication at 16hrs pi? The authors showed in figure 2 that their main effect of SPI2 and NO is earlier (maybe 8 hrs). It would be important for consistency to look at this. Timing is essential during infection and I believe the authors might find important information by looking at this.

Minor points:

Figure 1C/Table S1: At what time point the expression of these genes was measured? This is an important information to provide.

Figure 2A: It would be very informative to show the full panel of genes (as in Figure 1C). Why did the authors select only *ssaG* for this figure? Are all genes behaving similarly?

Does adding NO to the bacterial culture media (LB media) mimic what is observed during infection in term of induction of Spi2 components?

Does supplementing U937 cells with NO also enhance Spi2 induction and response to bacteria? This would be an important experiment to perform as it could be a good proof of concept.

A few studies discussed previously the link between Spi2 and PhoP/PhoQ but not in all context (PMID: 21625519, PMID: 15231802). This would be a good element of discussion.

There are a few variations in between figures as to which genes are used to look at Spi2 expression. The study would benefit with a bit consistency on that aspect as it would facilitate comparison between figures.

I could not find information about the Fnr ELISA in the method section. The authors should add this information and describe how they analyse the data. In Figure 5 H, They use N to describe the

concentration of *fnr* and I am not too sure what this means. How many times was this experiment performed? Figure 5 legend does not include this information for panel H.

The authors describe LDH in their method section but do not include this data in their manuscript

Do *Salmonella* promoter-replaced mutants replicate similarly to wt *Salmonella* in vitro? Do they infect macrophages as efficiently? As these are new strains, these experiments should be performed.

Spi2 has been shown to be important in human but differently than it is in mouse cells. The authors cannot claim it is less important (in their discussion, line 413-414). This study (PMID: 30368901) showed that *Spi2* subvert inflammasome activation in human primary cells and human cell lines (but not in mice).

Could the authors include in supplemental data an example of their Immunofluorescence staining (to support their graphs)

Reviewer #3 (Remarks to the Author):

The authors in this work investigate the effect of Nitric Oxide (NO) production on microbial burden in macrophages and mice infected with *Salmonella*. The data show that NO promotes bacterial replication in the early stages of the infection, while, in sharp contrast, it has completely the opposite effect during the later stages of infection. The latter is rather well known since early 2000, i.e. NO production via iNOS activation is a key host defence mechanism against *Salmonella*. The authors then move on to investigate the mechanism via which NO promotes early bacterial replication and present data to suggest that this depends on SPI-2 activation. Finally, they have also identified two bacterial signalling pathways that mediate SPI-2 activation.

I find the idea of NO promoting SPI-2 expression/activation entirely plausible. The pathogen has to survive in a very hostile intracellular environment and it is likely that early exposure to NO could "force" it to activate machinery required for its survival. Although microbial genetics are not my field of expertise, I do not have any major criticism for the work on SPI-2 activation and the signalling pathways mediating this effect.

However, I am yet to be convinced about NO promoting bacterial "replication" in macrophages and mice early during the infection and the biological significance of this observation. This is for a number of reasons which are given below.

First, I do not agree with the term "replication". To my opinion, the data does not show replication, it shows microbial burden/load or intracellular microbial numbers/counts. Have the authors actually observed in real time bacteria replicating/dividing during their experiments? They have only counted CFUs via plating or counted intracellular bacteria via microscopy. This is not evidence of bacterial replication. I expect the authors to replace "replication" in all graph labels and references in the text with "burden" or something similar that properly describes the data shown.

Second, the hallmark of *Salmonella* uptake by murine macrophages is inflammasome activation via NLR4. This drives very rapid pyroptotic cell death and IL-1 β /IL-18 secretion. I would expect a substantial number (around 50%) of primary or immortalised murine bone marrow-derived macrophages infected with wild-type *Salmonella* at an MOI of 10 for 30 min, with the bacteria centrifuged on the cells to promote uptake, to be lysed by 2h. If this is the case, then I cannot understand how the number of intracellular bacteria steadily increases from 2h to 16h in Fig 2E and 2G? LDH release data must be shown in order to see if cell death occurs under these specific

experimental conditions. I do acknowledge that there are 3 important deviations from common/standard approaches: bacteria have been grown to stationary phase (rather than log phase), bacteria have been opsonised (rarely seen in recent studies) and RAW cells (rather than BMDMs have been used). These may have an effect on the amount of cell death seen in this system. There is actually a section in Methods about LDH release assays using the Cytotox kit by Promega but such data is not at all shown in the manuscript. If cell death occurs, then the results from the intracellular microbial burden should be examined in the light of any differences in the number of cells remaining viable at later time points. Again, my experience and published data would suggest that wild-type macrophages infected with Salmonella will lyse very rapidly and cannot harbour more than 1-2 bacteria/cell, therefore I find it very odd that 12 hours after infection Fig 2G shows 10 bacteria/ wild-type cell (in average). The authors should provide microscopy images to support Fig 2G.

Third, I would like to know if the difference seen in microbial burden from the spleen of wild-type and iNOS KO mice during the early stages of infection is specific to the route of administration (i.p. in this case) or whether it is evident when other routes of systemic administration is used, such as i.v. Have the authors challenged the mice i.v. and, if not, I would like to suggest a small experiment in which 4-5 wild-type and 4-5 iNOS KO will be challenged i.v and spleen/liver collected for analysis at day 2.

Fourth, I do not really understand why the authors focus so much on the species-specific differences between murine and human cells shown in Fig 1 and also highlighted this topic in the abstract. The data is non-conclusive. They compare between two cell lines (rather than between primary cells which would have generated more reliable data) and found that the murine RAW cells generate more NO and, at the same time, have higher intracellular burden than human U937 cells. These two findings may be related but they also may be not. I understand that this is a positive correlation and that there may be some merit in this hypothesis but these data are far from conclusive. I cannot see how this element of species-specific differences in NO production adds further impact on this work and I would recommend to be removed from the manuscript.

Fifth, I definitely find the effect on NO production on SPI-2 activation interesting but I struggle to understand the biological significance of the increased microbial burden early when we all know that NO production is a required for restricting microbial spread in the tissues. I am under the impression that authors have not discussed this in their Discussion. Assuming that the authors address my comments and substantiate the validity of their observations, I would like to recommend to the authors and editor a change in the narrative and the story. The effect of NO on SPI-2 and the mechanisms mediating this should go first followed by the result of this effect on intracellular burden.

I finally have an issue of a different nature with this work. I do understand that different countries have different regulations when comes to animal experimentation, but I would strongly encourage the authors to stop performing "survival" assays and replace them with humane end point assays. There is no need for unnecessary animal suffering and we both know very well that a mouse that has lost 15-20% of each body weight, it is hunched and slower than its peers with piloerection and possibly ocular discharge has no chance of recovery and can be culled at this stage without compromising the quality of the data.

There is a discrepancy between the label in Fig 1B (CFU x 10⁵) and the figure legend (bacterial CFU/10⁵ macrophage cells).

Responses to reviewers

We thank the reviewers for their careful review. We have heeded their suggestions and have performed additional experiments to clarify and strengthen the findings. We believe, by doing so, we have improved and strengthened the manuscript substantially and we thank the reviewers for this. Our point-by-point responses to their specific comments are provided below.

Reviewer #1 (Remarks to the Author):

Here, Jiang et al present a comprehensive study to show that in mice, NO promotes replication of S. Typhimurium in the early stages of infection (i.e. up to day 5 post-infection in vivo and 12 hours post-infection in vitro) by activating the SPI-2 type III secretion system, a process known to facilitate intracellular survival of Salmonella enterica serovars. The authors further show that SPI-2 activation is mediated by Fnr and the PhoP/Q two-component regulatory system. This work would be of interest to the general Salmonella field as it is dependent on SPI-2 machinery. It may be of interest to those working on other intracellular bacterial pathogens that utilise a type 3 or 4 secretion system (e.g. Shigella, Legionella or Coxiella). It would be of specific interest to those in the Salmonella field who use both in vitro and in vivo models of infection to confirm mechanisms of virulence. The work is novel, but is limited to the mouse model of infection and outcomes relating to this. The work has been executed well, the statistical methods applied are sound, mouse numbers are sufficient in most figures. I have comments that would need to be addressed before the article could be accepted for publication, but overall, this is a valuable study for the field of bacterial pathogenesis and adds insight into bacterial mechanisms of disease and also host resistance to infection.

Major comments:

1. In the abstract: The authors overstate the application of the study a little (new pathway, important, etc, but it's only in mice) – these novel observations in mice should be published because they are not relevant to the human disease model, and so should

be taken into consideration when researchers are making conclusions and assumptions about disease outcomes for humans when using the mouse model of Salmonella infection.

Response:

Thank you for the comment. We have modified the abstract to avoid overstating the application of the study.

“This study reveals a novel host signaling-mediated virulence activation pathway in *S. Typhimurium*” in the abstract has been modified as “This study reveals a host signaling-mediated virulence activation pathway in *S. Typhimurium*”. “providing important insights into *Salmonella* pathogenesis and host–pathogen interaction” has been modified as “providing further insights into *Salmonella* pathogenesis and host–pathogen interaction”. “It is likely that this pathway is not activated in human macrophages, owing to the lack of NO production.” has been deleted.

2. In the introduction: I am not a great believer in the idea that mouse infection with STm is a robust model for human Typhi infection, as each bacterium has a significantly different genetic makeup. This paper should probably make the argument that it could draw parallels in host responses during human Typhi infection and mouse STm infection so that it MAY be relied upon as a model for Typhi or maybe better still, invasive STm serovars. This is currently not clear in the introduction.

Response:

Thank you for the comment. We agree with the reviewer that the mouse infection with *S. Typhimurium* may be not a robust model for human *S. Typhi* infection. “Owing to the lack of suitable human models, mouse model of *S. Typhimurium* infection has long been used as an established model for studying the pathogenesis of human typhoid fever induced by human-restricted serovars Typhi and Paratyphi ” has been deleted from the introduction section.

Possible impact of NO production on invasive *S. Typhimurium* ST313 is discussed (lines 414–416).

3. Do the authors see similar dynamics of NO production in both RAW264.7 and U937

with heat-killed STm and /or LPS treatment? This is another way of testing the involvement of the SPI-2 T3SS components.

Response:

Thank you for the comment. We have now determined the NO production in RAW264.7 and U937 cells with LPS treatment. NO production was also higher in RAW264.7 cells than in U937 cells as determined at 24 h post-treatment (Supplementary Fig. 1b), in agreement to the results obtained with *S. Typhimurium* infection. The result was described in the revised manuscript (lines 123–124).

4. The cell lines compared (RAW264.7 vs U937) have some very innate differences that can influence the replication dynamics. RAW264.7 cells are missing components of the inflammasome, a critical innate detection and response mechanism against intracellular pathogens. Do the authors have any data from mouse bone-marrow-derived macrophages to support the replication dynamics they observe in RAW264.7 cells over the chosen infection time course – this could be placed as a supplemental figure. How can the authors determine that the lack of control of STm is not due to the lack of inflammasome activation (which would be occurring in U937 cells).

Response:

Thank you for the comment. We have now determined the replication dynamics of *S. Typhimurium* in mouse bone-marrow-derived macrophages (BMDMs) and human peripheral blood mononuclear cells (PBMCs). We found that intracellular bacterial burden of *S. Typhimurium* was higher in mouse BMDMs than in human PBMCs (Supplementary Fig. 1d), confirming the correlation between NO production and bacterial growth in mouse macrophages. However, better growth of *S. Typhimurium* in RAW264.7 cells (17.4-fold at 16 h) than in BMDMs (3.2-fold at 16 h) was detected, which is likely due to the lack of the inflammasome activation in RAW264.7 cells. The results were described in the revised manuscript (lines 129–134).

5. In Fig 2D, there is no explanation for why 16 hrs post-infection was selected, was this just as a mid-infection test or did other time points show similar results?

Response:

At 16 h of infection, high intracellular bacterial burden and high NO production while the death of cells was not significant (Supplementary Fig. 3b). Therefore, we selected 16 hpi to assess the effect of supplementing iNOS^{-/-} cells with NO on *ssaG* expression. We have now further assessed the effect at 4, 8, and 24 hpi. The results showed that the supplementation of NO increased *ssaG* expression in iNOS^{-/-} PMs at all the tested time points (Fig. 2d). This information has been added to the revised manuscript (lines 238–240).

6. Line 209: why ip infection rather than physiological oral infection? You will still see dissemination to peripheral organs via oral infection route so not sure why only ip was used?

Response:

i.p. injection has been widely used to investigate the systemic infection of *Salmonella*, as it allows *Salmonella* to directly disseminate to the systemic sites, bypassing the need for invasion of the intestine as in oral infection, which would be affected by the difference in host mucosal immunity and bacterial invasion ability. Here, we aimed to investigate the contribution of host NO production to *S. Typhimurium* systemic infection, thus we used i.p. injection and then compared the bacterial burden in the liver and spleen of the infected C57BL/6 wild-type and iNOS^{-/-} mice.

7. Does the P_{yfiR}::P_{ssrA} strain induce less nitrite in vitro? (as per assay in Fig 1A?) This would help support their conclusions on lines 257-260.

Response:

Thank for the comment. We have now determined the production of NO in P_{yfiR}::P_{ssrA}-infected RAW264.7 cells and wild-type-infected RAW264.7 cells, at 0, 8, 16, and 24 hpi. The results showed similar NO production in P_{yfiR}::P_{ssrA}-infected and wild-type-infected RAW264.7 cells (Supplementary Fig. 3e), indicating that P_{yfiR}::P_{ssrA} mutation does not influence macrophage NO production, in support of that reduced virulence of P_{yfiR}::P_{ssrA} is associated with the reduced SPI-2 expression. This information has been added to the revised manuscript (lines 272–273).

8. *Fig 4A does not indicate at what time point this analysis was performed? Fig 4B indicates analyses were performed at 16 hpi, was this the same for 4A? And what about PMs from C57BL/6 mice? Do the authors see similar results in PMs?*

Response:

Thank you for pointing it out. The analysis presented in Fig. 4a was performed at 16 hpi, and this has now been added to the figure 4a legend. We have now performed the same analysis with isolated PMs and found similar results as that with RAW264.7 cells (Fig. 4a right panel and Fig. 4b right panel). The results have been described in the revised manuscript (lines 287, 291).

9. *The authors used 2 mouse macrophage cell lines to show results, but only one human macrophage cell line. Do the authors have any data in THP-1 macrophages or human PBMCs to reduce the likelihood of a cell specific effect? I don't expect to see all results repeated in another cell line, but one significant result showing lack of NO production in another human cell line would be appreciated and increase the robustness of the results.*

Response:

Thank you for the comment. We have now investigated the NO production in THP-1 macrophages and human PBMCs. Up to 2 and 3 μ M NO, respectively, was detected during the 24 h infection period (Supplementary Fig. 1a,c), further confirming the negligible NO production in human macrophages. This information has been added to the revised manuscript (lines 122, 129–131).

10. *Can the authors comment on whether the NO pathway is active in other species that are major reservoirs for Salmonella enterica, e.g. pigs, cows? This may add some weight to the relevance of the findings.*

Response:

Thank you for the suggestion. We have added a paragraph to discuss whether the NO pathway is active in other species in the Discussion section of the revised manuscript (lines 442–452).

Minor comments:

1. Figure 1C: legend has no details as to the type RNA-seq undertaken at what time point of infection the RNA was processed. Supp Fig 1 should be amended accordingly.

Response:

The relevant information was now added to Fig. 1c and Supplementary Fig. 1e of the revised manuscript.

2. Fig 2A, B and C: keep y axis the same, i.e. all max at 30 – it is more transparent, otherwise the reader could assume similar levels of overall expression at first glance. Same for Fig 2E and F, maybe even Fig 2H and I also.

Response:

We have revised the Fig. 2 as suggested.

3. The strain name of the STm being used should be mentioned in both the methods and in the results (just once at the start – not all the way through) or at least in the figure legends.

Response:

The strain name of *S. Typhimurium* has now been mentioned in both the methods (line 467) and in the results (line 121), and also in Fig. 1a legend.

4. In Fig 2J, it is not immediately clear that this is an ssaB mutant being used. I suggest including a reference to it in the figure (i.e. on the y axis legend, say fold intracellular replication of ssaB).

Response:

Thank you for pointing it out. As suggested, we have revised the y axis legend of this figure as “Increase in Δ ssaB burden (fold relative to 2 hpi)”.

5. In Fig 2G – no details for what stain was used to count bacteria. Regardless of whether these details are in the methods or not – there should be some indication of how this was done withing the figure or the results text so that the reader can immediately decide whether this was a reliable/viable/robust assessment.

Response:

We have added the information in the Fig. 2g legend.

6. Line 43: add 'the' in front of 'mouse model of *S. Typhimurium*... '.

Response:

Done.

7. Line 46: change the word 'aspect' to 'hand'

Response:

Done.

8. Line 56: Add 'The' before 'SPI-2-encoded...'

Response:

Done.

9. Line 57: change to 'outside of the SPI-2 operon, ...'

Response:

Done.

10. Line 58: add 'the' before 'Salmonella-containing vacuole....'

Response:

Done.

11. Line 66: add 'observed' before 'in mouse macrophages'

Response:

Done.

12. Lines 85 and 86: add 'the' before 'transcriptional level' and before 'posttranscriptional level'.

Response:

Done.

13. Line 104: change start of sentence to “In this study, the contribution of host NO production to *S. Typhimurium* infection in mice was investigated using ...’

Response:

Done.

14. Line 115: add ‘the’ before ‘innate immune...’

Response:

Done.

15. Line 141: change ‘modest’ to ‘modestly’

Response:

Done.

16. Line 158: Fig 2D reference is in bold while others are not.

Response:

Done.

17. Line 222: change ‘indicated’ to ‘indicate’

Response:

Done.

18. Line 241: add ‘the’ after the word ‘that’

Response:

Done.

19. Line 305: change ‘a’ to ‘an’

Response:

Done.

20. Line 305: change ‘level’ to ‘levels’

Response:

Done.

21. Line 315: change 'consisting' to 'consistent'

Response:

Done.

22. Line 357: add the word 'the' before 'host innate immune'

Response:

Done.

23. Line 361: change 'level' to 'levels'

Response:

Done.

24. Line 365: remove the word 'an'

Response:

Done.

25. Line 370: add 'the' before 'SCV'

Response:

Done.

26. Line 389: change 'report' to 'reports'

Response:

Done.

27. Line 405: remove the word 'of'

Response:

Done.

28. Line 411: remove the word 'the'

Response:

Done.

29. Line 423: add 'the' before 'host innate immune'

Response:

Done.

30. Line 468: change 'mouse' to 'mice'

Response:

Done.

Reviewer #2 (Remarks to the Author):

In their manuscript, Jiang and colleagues explore the role of nitric oxide as an innate immune cue during Salmonella infection. They suggest that production of nitric oxide in murine macrophages (but not in human macrophages) leads to Spi2 induction and promote Salmonella replication at early time-point. Their results also show that this is the opposite at later time points, and that NO is required to control infections. Their data suggest that PhoP and Fnr regulate Spi2-related genes expression in response to NO.

Overall the observations presented in the paper are interesting and will be of interest to the field. They provide interesting genetic approaches to validate their hypothesis. However, I feel there is some caveats that will need to be fixed before acceptance of this publication. I hope my comments will be useful to the authors.

Major points:

1. Figure 1: The claim that mouse but not human induce NO during Salmonella infection is based on the usage of cell lines. Although these are good resources, conclusions cannot be drawn as they do not behave in a similar manner as primary cells. For example, RAW cells do not express competent inflammasome (lack ASC/PYCARD) whereas U937 express competent inflammasome component. This should be tested in primary macrophages. It has been shown by some studies (PMID: 23774601) that primary human macrophages can produce NO. The authors should be careful about drawing strong conclusion from cell lines.

Response:

Thank you for the comment. We have now determined the replication dynamics of *S. Typhimurium* in mouse bone-marrow-derived macrophages (BMDMs) and human peripheral blood mononuclear cells (PBMCs). We found that intracellular bacterial burden of *S. Typhimurium* was higher in mouse BMDMs than in human PBMCs (Supplementary Fig. 1d), confirming the correlation between NO production and bacterial growth in mouse macrophages. The results were described in the revised

manuscript (lines 129–131).

Although NO production was detected in some human primary macrophages in previous studies and in this study, but the level is very low, which is insufficient to induce SPI-2 expression. For better clarification, we have reworded the description of NO production in human macrophages throughout the revised manuscript (lines 75–76, 117, 406–411), and cited the paper (*PMID: 23774601*).

2. The authors used a bunch of Salmonella mutants (various promoters to control Spi2 expression). Their approach is certainly interesting but does not fully imply that Spi2 is the main element responsible here. Why not simply include an ssaR ko strain. This strain is commonly used as a Spi2 inactive mutants and would be an important control here. This would be a good control for their infection model (although I would understand this may represent too much work).

Response:

Thank you for the comment. In this work, we used a *ssrB* mutant (Δ *ssrB*) to inactivate SPI-2 gene expression, based on that SsrA-SsrB is the essential regulatory system required for the expression of all SPI-2 genes. We also confirmed that the expression of five representative SPI-2 genes (*ssaG*, *sifA*, *sipC*, *ssaE*, *sscA*, *ssaV*) was eliminated in Δ *ssrB* (data not shown) and that Δ *ssrB* did not replicate in untreated RAW264.7 cells (Fig. 2j). Therefore, Δ *ssrB* was also suitable for this study. In future studies, we will consider *ssaR* ko strain for SPI-2 inactivation.

3. Figure 3B/C: Why looking at the replication at 16hrs pi? The authors showed in figure 2 that their main effect of SPI2 and NO is earlier (maybe 8 hrs). It would be important for consistency to look at this. Timing is essential during infection and I believe the authors might find important information by looking at this.

Response:

At 16 h of infection, high intracellular bacterial burden and high NO production while the death of cells was not significant (Supplementary Fig. 3b). Therefore, we selected 16 hpi to assess the replication ability of different SPI-2-expressing derivatives and the effect of supplementing iNOS^{-/-} cells with NO on *ssaG* expression. This

information has been added to the revised manuscript (lines 238–240).

Minor points:

1. Figure 1C/Table S1: At what time point the expression of these genes was measured? This is an important information to provide.

Response:

Thank you for pointing it out. The bacteria used for RNA-seq and qRT-PCR were collected at 8 h post-infection. The information was now added in Supplementary Table 1 and the legend of Fig. 1.

2. Figure 2A: It would be very informative to show the full pannel of genes (as in Figure 1C). Why did the authors select only ssaG for this figure? Are all genes behaving similarly?

Response:

During *S. Typhimurim* growth within host cells, the expression of SPI-2 genes is under the control of the SPI-2-encoded regulator system SsrA-SsrB (*PMID: 19264535*). Based on initial RNA-seq profiling and qRT-PCR analysis of 7 representative SPI-2 genes, the expression of all SPI-2 genes was induced *via ssrA/B* in RAW264.7 cells (Supplementary Fig. 1e). Although it is more informative to show the full panel of genes, the use of *ssaG* as a representative to monitor the expression of SPI-2 genes in mouse macrophage was also sufficient in this study (Fig. 2).

3. Does adding NO to the bacterial culture media (LB media) mimic what is observed during infection in term of induction of Spi2 components?

Response:

We used N-minimal medium that mimic the low Mg^{2+} and low phosphorus conditions of the macrophages to assess the effect of NO (0 to 200 μM) on SPI-2 expression (*ssaG*). The results showed that *ssaG* expression was increased by the addition of 10–20 μM NO, under low O_2 condition (a condition in macrophages) (Fig. 5g), in agreement to NO levels (5–20 μM) detected for SPI-2 induction in RAW264.7

cells. This is described in the result section (lines 326–329).

4. Does supplementing U937 cells with NO also enhance Spi2 induction and response to bacteria? This would be an important experiment to perform as it could be a good proof of concept.

Response:

Thank you for the comment. We have now performed the experiment to investigate the effect of supplementing U937 cells with NO on the intracellular expression of SPI-2 *ssaG* gene. The result showed that the supplementation of 20 μ M NO increased *ssaG* expression, ~1.8-fold, in U937 at 16 h post-infection (Supplementary Fig. 5a). We have discussed this result in the revised manuscript (lines 423–424).

5. A few studies discussed previously the link between Spi2 and PhoP/PhoQ but not in all context (PMID: 21625519, PMID: 15231802). This would be a good element of discussion.

Response:

Thank you for the comment. We have now cited these papers and discussed the possible link between these studies and our work here in the revised manuscript (lines 376–383).

6. There are a few variations in between figures as to which genes are used to look at Spi2 expression. The study would benefit with a bit consistency on that aspect as it would facilitate comparison between figures.

Response:

Thank you for the comment. *ssaG* was used as a representative to monitor the expression of SPI-2 genes in all qRT-PCR experiments. Four SPI-2 representative genes (*ssaE*, *sscA*, *ssaG*, and *sifA*) were used to monitor the expression of SPI-2 genes in *ssrA* promoter replacement derivatives during infection of RAW264.7 cells. To improve clarity, we have now revised the description of this section (lines 232–236). In future works, we will be more carefully to select SPI-2 representative genes for consistency.

7. I could not find information about the Fnr ELISA in the method section. The authors should add this information and describe how they analyse the data. In Figure 5 H, They use *N* to describe the concentration of fnr and I am not too sure what this means. How many time was this experiment performed? Figure 5 legend does not include this information for panel H.

Response:

Thank you for the comment. The Fnr ELISA protocol have now been added in the method section (lines 660–674). In Figure 5H, *N* indicates the molecular number of Fnr protein. For each sample, the number of Fnr moleculars per bacterium was calculated based on the number of bacterial cells as determined by plate count, the concentration of Fnr proteins as determined by ELISA, and molar mass of Fnr and the reaction between NO and [4Fe-4S] of Fnr (each [4Fe-4S] cluster of Fnr reacts with maximum eight NO molecules). Data are obtained from three independent experiments. We have added the information in Fig. 5h legend.

8. The authors describe LDH in their method section but do not include this data in their manuscript

Response:

Thank you for pointing it out. In this work, we assessed the mortality of RAW264.7 cells after infection with *S. Typhimurium* wild-type at 8, 16, and 24 hpi, by measuring LDH release, but the data was not presented in the original manuscript. We have now presented the result in the revised manuscript (line 240).

9. Do *Salmonella* promotor-replaced mutants replicate similarly to wt *Salmonella* in vitro? Do they infect macrophages as efficiently? As these are new strains, these experiments should be performed.

Response:

Thank you for the comment. We have now determined the growth of 4 lower-SPI-2-expressing derivatives ($P_{I150}::P_{ssrA}$, $P_{sinR}::P_{ssrA}$, $P_{yfiR}::P_{ssrA}$, and $P_{aceB}::P_{ssrA}$) *in vitro*. We observed that all of these derivatives grew as well as wild-type in LB medium (Supplementary Fig. 3c). We have also found that these derivatives strains infected

RAW264.7 cells as efficiently as wild-type (Supplementary Fig. 3d). The results are now presented in the revised manuscript (lines 246–250).

10. Spi2 has been shown to be important in human but differently that it is in mouse cells. The authors cannot claim it is less important (in their discussion, line 413-414). This studie (PMID: 30368901) showed that Spi2 subvert inflammasome activation in human primary cells and human cell lines (but not in mice).

Response:

Thank you for the comment. We have now cited this paper and modified relevant content in the Discussion accordingly (lines 419–423).

11. Could the authors include in supplemental data an example of their Immunofluorescence staining (to support their graphs)

Response:

Representative immunofluorescence images (8, 12, and 24 hpi) are now provided in the revised manuscript (Supplementary Fig. 2b).

Reviewer #3 (Remarks to the Author):

The authors in this work investigate the effect of Nitric Oxide (NO) production on microbial burden in macrophages and mice infected with Salmonella. The data show that NO promotes bacterial replication in the early stages of the infection, while, in sharp contrast, it has completely the opposite effect during the later stages of infection. The latter is rather well known since early 2000, i.e. NO production via iNOS activation is a key host defence mechanism against Salmonella. The authors then move on to investigate the mechanism via which NO promotes early bacterial replication and present data to suggest that this depends on SPI-2 activation. Finally, they have also identified two bacterial signalling pathways that mediate SPI-2 activation.

I find the idea of NO promoting SPI-2 expression/activation entirely plausible. The pathogen has to survive in a very hostile intracellular environment and it is likely that early exposure to NO could “force” it to activate machinery required for its survival. Although microbial genetics are not my field of expertise, I do not have any major criticism for the work on SPI-2 activation and the signalling pathways mediating this effect.

However, I am yet to be convinced about NO promoting bacterial “replication” in macrophages and mice early during the infection and the biological significance of this observation. This is for a number of reasons which are given below.

1. First, I do not agree with the term “replication”. To my opinion, the data does not show replication, it shows microbial burden/load or intracellular microbial numbers/counts. Have the authors actually observed in real time bacteria replicating/dividing during their experiments? They have only counted CFUs via plating or counted intracellular bacteria via microscopy. This is not evidence of bacterial replication. I expect the authors to replace “replication” in all graph labels and references in the text with “burden” or something similar that properly describes the data shown.

Response:

Thank you for the comment. “Relative fold replication” have been replaced by “Increase in bacterial burden” in all graph labels. “Replication ability” have been replaced by “bacterial burden” or “intracellular bacterial burden” in the text.

2. Second, the hallmark of Salmonella uptake by murine macrophages is inflammasome activation via NLRC4. This drives very rapid pyroptotic cell death and IL-1b/IL-18 secretion. I would expect a substantial number (around 50%) of primary or immortalised murine bone marrow-derived macrophages infected with wild-type Salmonella at an MOI of 10 for 30 min, with the bacteria centrifuged on the cells to promote uptake, to be lysed by 2h. If this is the case, then I cannot understand how the number of intracellular bacteria steadily increases from 2h to 16h in Fig 2E and 2G? LDH release data must be shown in order to see if cell death occurs under these specific experimental conditions. I do acknowledge that there are 3 important deviations from common/standard approaches: bacteria have been grown to stationary phase (rather than log phase), bacteria have been opsonised (rarely seen in recent studies) and RAW cells (rather than BMDMs have been used). These may have an effect on the amount of cell death seen in this system. There is actually a section in Methods about LDH release assays using the Cytotox kit by Promega but such data is not at all shown in the manuscript. If cell death occurs, then the results from the intracellular microbial burden should be examined in the light of any differences in the number of cells remaining viable at later time points. Again, my experience and published data would suggest that wild-type macrophages infected with Salmonella will lyse very rapidly and cannot harbour more than 1-2 bacteria/cell, therefore I find it very odd that 12 hours after infection Fig 2G shows 10 bacteria/ wild-type cell (in average). The authors should provide microscopy images to support Fig 2G.

Response:

Thank you for the comment. In this work, we have designed our macrophage infection assays to reduce cell death. To prevent SPI-1-induced rapid macrophage death, cells were infected with *S. Typhimurim* grown to late stationary phase when the SPI-1 TTSS is not expressed; and the bacteria have been opsonised to increase the active phagocytosis of the bacteria by macrophages. We assessed the mortality of RAW264.7

cells after infection with *S. Typhimurium* wild-type at 8, 16, and 24 hpi, by measuring LDH release, but the data was not provided in the original manuscript. We found that at 16 h of infection, the death of cells was not significant (Supplementary Fig. 3b). Therefore, we selected 16 hpi to assess the replication ability of different SPI-2-expressing derivatives and the effect of supplementing iNOS^{-/-} cells with NO on *ssaG* expression. We have now added the results in the revised manuscript (lines 238–240). Considering that the difference in cell number would affect the results of bacterial growth, when calculate the increase in intracellular bacterial burden, the bacterial CFUs have been normalized by the numbers of living cells. We have added the information in the revised manuscript (lines 573–575).

It has been reported that during the stage of rapid growth of *Salmonella* in mouse liver and spleen, the majority of infected phagocytes contain relatively few bacteria (PMID: 17931727). The relatively low number of bacteria in each infected macrophage *in vivo* might be associated with the interaction between macrophages and other immune cells (PMID: 19079353). However, the results obtained from *in vitro* cell culture models by our group and other groups have showed that a proportion of the infected macrophages contain more than 10 bacteria (PMID: 21376231). Please also see the Figure below.

***S. Typhimurium* infection of RAW264.7 cells at 24 hpi (Green: *Salmonella*; Blue, nucleus)**

Representative immunofluorescence images (8, 12, and 24 hpi) are now provided in the revised manuscript (Supplementary Fig. 2b).

3. Third, I would like to know if the difference seen in microbial burden from the spleen of wild-type and iNOS KO mice during the early stages of infection is specific to the

route of administration (i.p. in this case) or whether it is evident when other routes of systemic administration is used, such as i.v. Have the authors challenged the mice i.v. and, if not, I would like to suggest a small experiment in which 4-5 wild-type and 4-5 iNOS KO will be challenged i.v and spleen/liver collected for analysis at day 2.

Response:

Thank you for the comment. We have now performed the *i.v.* experiment as suggested. The results showed that WT mice had higher bacterial burden in their liver and spleen than the *iNOS*^{-/-} mice at day 2 post-infection (Supplementary Fig. 2d), which is consistent with the *i.p.* results, indicating that the NO-induced virulence promotion at the early infection stage is not associated with the infection route. The new results are now described in the revised manuscript (lines 211–214).

4. Fourth, I do not really understand why the authors focus so much on the species-specific differences between murine and human cells shown in Fig 1 and also highlighted this topic in the abstract. The data is non-conclusive. They compare between two cell lines (rather than between primary cells which would have generated more reliable data) and found that the murine RAW cells generate more NO and, at the same time, have higher intracellular burden than human U937 cells. These two findings may be related but they also may be not. I understand that this is a positive correlation and that there may be some merit in this hypothesis but these data are far from conclusive. I cannot see how this element of species-specific differences in NO production adds further impact on this work and I would recommend to be removed from the manuscript.

Response:

Thank you for the comment. We agree with that the species-specific differences between mouse and human macrophages were not the focus of this study. “It is likely that this pathway is not activated in human macrophages, owing to the lack of NO production.” was deleted from the abstract.

The results for the species-specific differences between mouse and human macrophages (NO production, bacterial burden, SPI-2 gene expression) are presented to support the conclusion made for the NO-dependent *S. Typhimurium* virulence

mechanism in mice (Fig. 1a-c; Fig. 3a). We think it is better to present those data in the Result section. In response to the comment by Reviewer #1, we have also determined the NO production and *S. Typhimurium* growth in primary mouse bone-marrow-derived macrophages (BMDMs) and primary human peripheral blood mononuclear cells (PBMCs), and the result is presented in the revised manuscript (Supplementary Fig. 1c, d; lines 129–131).

5. Fifth, I definitely find the effect on NO production on SPI-2 activation interesting but I struggle to understand the biological significance of the increased microbial burden early when we all know that NO production is a required for restricting microbial spread in the tissues. I am under the impression that authors have not discussed this in their Discussion. Assuming that the authors address my comments and substantiate the validity of their observations, I would like to recommend to the authors and editor a change in the narrative and the story. The effect of NO on SPI-2 and the mechanisms mediating this should go first followed by the result of this effect on intracellular burden.

Response:

Thank you for the comment. We have added a section to discuss “the biological significance of the increased microbial burden at the early infection stage” (lines 432–441).

6. I finally have an issue of a different nature with this work. I do understand that different countries have different regulations when comes to animal experimentation, but I would strongly encourage the authors to stop performing “survival” assays and replace them with humane end point assays. There is no need for unnecessary animal suffering and we both know very well that a mouse that has lost 15-20% of each body weight, it is hunched and slower than its peers with piloerection and possibly ocular discharge has no chance of recovery and can be culled at this stage without compromising the quality of the data.

Response:

Thank you for the comment. We agree with the reviewer that there is no need for unnecessary animal suffering. We will consider the suggestions in our future work.

7. *There is a discrepancy between the label in Fig 1B (CFU x 10⁵) and the figure legend (bacterial CFU/10⁵ macrophage cells).*

Response:

Thank you for pointing it out. We have revised the Figure legend (line 1045).

Other changes:

1. The method used to isolate and culture mouse BMDMs has now been added to the Materials and Methods section of the revised manuscript (lines 550–554).
2. The “Data availability” and “Statistical analysis” sections has been revised according to the requirement of *Communications Biology* (lines 700–708).
3. All bar graphs in the original manuscript have been replaced with plots that feature information about the distribution of the underlying data.

Reviewers' comments:

Reviewer #1 (Remarks to the Author):

Thank you for addressing my comments comprehensively, I am happy for this work to be published.

Reviewer #2 (Remarks to the Author):

I thank the authors for carefully addressing most of my comments. I have a few more concerns to raise from their comments:

Minor concerns:

1- Supplementary figure 1: How many donors did they authors use for their PBMC experiments? The error bar on the Nitrite graph is absent, which I find surprising considering the usual high variability observed with human cells.

2- The authors should acknowledge that RAW cells do not express ASC/Pycard.

3- Data showing that ssrV eliminate the expression of Spi2 genes should be displayed as supplemental, especially as the authors claimed they did the experiment.

4- Figure 3B/C: Why looking at the replication at 16hrs pi? The authors showed in figure 2 that their main effect of SPi2 and NO is earlier (maybe 8 hrs). It would be important for consistency to look at this. Timing is essential during infection, and I believe the authors might find important information by looking at this. I mentioned this element before, but I am not satisfied with the authors' answer here. Why not including both time points? The paper is not uniform in terms of time-point, and this is an element of concern to me.

Responses to Reviewer 2

Remarks to the Author:

I thank the authors for carefully addressing most of my comments. I have a few more concern to raise from their comments:

Response:

Thank you for your consideration of this work. We appreciate your effort in helping us improve our manuscript. We have carefully revised the manuscript based on your comments and suggestions. Detailed point-by-point responses are provided below.

Minor concerns:

1. Supplementary figure 1: How many donors did they authors use for their PBMC experiments? The errors bar on the Nitrite graph is absent, which I find surprising considering the usual high variability observed with human cells.

Response:

Thank you for the comment. Human peripheral blood CD14⁺ monocytes (PBMCs) used in this study were not isolated from the peripheral blood of different donors, but obtained from STEMCELL Technologies (1×10^7 cells under the same batch) (Catalog#70035; <https://www.stemcell.com/human-peripheral-blood-cd14-monocytes-frozen.html>). We added the website in the revised manuscript (line 536).

2. The authors should acknowledge that RAW cells do not express ASC/Pycard.

Response:

Thank you for the comment. We have added this information in the revised manuscript (lines 134–135).

“The increase in bacterial burden was less in BMDMs (3.2-fold at 16 h) than in RAW264.7 cells (17.4-fold at 16 h), and this was likely due to the absence of inflammasome activation in RAW264.7 cells, **which do not express the inflammasome adapter protein ASC.**”

3. Data showing that *ssrV* eliminate the expression of Spi2 genes should be display as supplemental, especially as the authors claimed they did the experiment.

Response:

The result that mutation of *ssrB* eliminated the expression of six representative SPI-2 genes (*ssaG*, *sifA*, *sipC*, *ssaE*, *sscA*, *ssaV*) is now displayed in Supplementary figure 2c.

Supplementary Fig. 2c, qRT-PCR analysis of *ssaG*, *sifA*, *sipC*, *ssaE*, *sscA*, and *ssaV* mRNA levels in *S. Typhimurium* wild-type and *ssrB* mutant. RNA was extracted from bacteria grown in N-minimal medium. Data are presented as mean \pm SD, n = 3 independent experiments.

4. Figure 3B/C: Why looking at the replication at 16hrs pi? The authors showed in figure 2 that their main effect of SPI2 and NO is earlier (maybe 8 hrs). It would be important for consistency to look at this. Timing is essential during infection, and I believe the authors might find important information by looking at this. I mentioned this element before, but I am not satisfied with the authors answer here. Why not including both time point? The paper is not uniform in term of time-point, and this is an element of concern to me.

Response:

S. Typhimurium starts to replicate in mouse macrophages from 4 hpi, and both 8 and 16 hpi are at the active replication stage (Fig. 1c and Supplementary Fig. 1d). The effect of NO production on SPI-2 expression was demonstrated at 8 and 16 hpi (Fig. 2a-d), suggesting the effect of NO on replication at both time points. We selected 16 hpi to test the replication difference between wild-type and mutant strains, as that the difference at this time point might be more significant due to the higher intracellular bacterial burden at 16 hpi relative to 8 hpi. We agree with the reviewer that it is more

informative to test both time points. We will design our experiments more carefully for consistency in the future.

REVIEWERS' COMMENTS:

Reviewer #2 (Remarks to the Author):

The authors have addressed my concerns

Responses to Reviewer 2

Remarks to the Author:

The authors have addressed my concerns

Response:

Thank you for your consideration of this work. We appreciate your effort in helping us improve our manuscript.